# FRUSTRATINGLY EASY MODEL GENERALIZATION BY DUMMY RISK MINIMIZATION

## ABSTRACT

Empirical risk minimization (ERM) is a fundamental machine learning paradigm. However, its generalization ability is limited in various tasks. In this paper, we devise Dummy Risk Minimization (`DuRM`), a frustratingly easy and general technique to improve the generalization of ERM. `DuRM` is extremely simple to implement: just enlarging the dimension of the output logits and then optimising using standard gradient descent. Moreover, we validate the efficacy of `DuRM` on both theoretical and empirical analysis. Theoretically, we show that `DuRM` derives greater variance of the gradient, which facilitates model generalization by observing better flat local minima. Empirically, we conduct evaluations of `DuRM` across different datasets, modalities, and network architectures on diverse tasks, including conventional classification, semantic segmentation, out-of-distribution generalization, adversarial training, and long-tailed recognition. Results demonstrate that `DuRM` could consistently improve the performance under all tasks with an almost free-lunch manner. The goal of `DuRM` is not achieving state-of-the-art performance, but triggering new interest in the fundamental research on risk minimization.

## 1 INTRODUCTION

Deep learning has demonstrated remarkable achievements across diverse fields, such as image classification (Deng et al., 2009; He et al., 2016; Vaswani et al., 2017; Radford et al., 2021), semantic segmentation (Long et al., 2015; Xie et al., 2021; Cordts et al., 2016; Everingham et al., 2015), speech recognition (Baevski et al., 2021; Schneider et al., 2019), and natural language processing (Vaswani et al., 2017; Devlin et al., 2018). In machine learning community, empirical risk minimization (ERM) (Sain, 1996) serves as the fundamental paradigm, in which various algorithms are developed to enhance *generalization* performance in various scenarios based on it, including out-of-distribution (OOD) generalization (Wang et al., 2022), long-tailed recognition (Tang et al., 2020), and adversarial defense (Goodfellow et al., 2015; Carlini & Wagner, 2017).

To enhance model generalization, existing efforts employ various strategies, either by incorporating versatile modules into ERM or by designing modules tailored to specific settings. On one hand, general techniques with broad effectiveness across different tasks are applied to ERM through regularization methods (e.g., $\ell_1$ or $\ell_2$), data augmentation (Cubuk et al., 2020), and ensemble learning (Freund & Schapire, 1997). On the other hand, researchers developed specific modules to improve generalization in particular tasks. For example, invariant risk minimization (IRM) (Arjovsky et al., 2019) enhances OOD generalization by learning class label related causal features. Adversarial training (Madry et al., 2017) improves adversarial robustness, while weight balancing techniques (Yang & Xu, 2020) successfully achieved better performance in long-tailed recognition.

Albeit that most efforts show great performance, their complexity cannot be ignored. Especially when the training data and network architectures become larger, ERM still remains a strong solution in most applications due to its simplicity. For example, Gulrajani & Lopez-Paz (2020) claimed that current algorithms do not significantly outperform ERM in OOD generalization. Hence, how to develop a general, simple, and effective improvement to ERM for better generalization remains a major challenge.

To achieve better improvement with exploiting failure attributes of ERM, SWA (Izmailov et al., 2018) proved that more flat local minima could benefit model generalization via better convergence, while ERM-motivated model convergence is suspended at the tipping point of a flat landscape

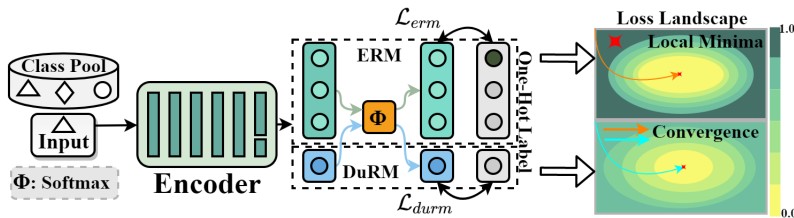

Figure 1: Illustration of `DuRM`, which adds dummy classes to the output layer while keeping the ground-truth labels unchanged. We show that `DuRM` makes model converge toward a more flat local minima than ERM theoretically (Sec. 2) and empirically (Sec. 3).

without entering the optimal point. Thus, the generalization of ERM is inadequate, especially in scenarios involving outliers, *i.e.*, dense classification, domain shift, and adversarial attacking. In these scenarios, the existence of outliers leads to increased uncertainty and differs the landscapes enormously between training and testing distributions (Cha et al., 2021), resulting in unsatisfactory generalization performance of ERM.

In this paper, we devise a frustratingly easy paradigm called **Du**mmy **R**isk **M**inimization (`DuRM`, Figure 1) to improve ERM's generalization ability. Concretely, *DuRM enlarges the dimension of output logits for better generalization*, which is inspired by the discussion (LAWRENCE, 1996) about the number of hidden nodes in deep networks (Zagoruyko & Komodakis, 2016; Szegedy et al., 2017; Tan & Le, 2019), where better generalization is achieved by enhancing the width of intermediate layers. `DuRM` is extremely easy to implement: just adding additional dimensions (which we call *dummy class*) to the output logits. For instance, we design 12 neurons instead of 10 in the output layer of the neural network for CIFAR-10 (Krizhevsky et al., 2009) classification. Different from adding width for intermediate layers which magnifies parameter scale to better capture latent information, expanding logits provides implicit supervision for existing classes, thus facilitating model optimization. This can also be understood as increasing the degree of freedom to the classifier.

Theoretically, we first prove that dummy classes only provide additional gradient risk and *no* samples can be classified to them. We then demonstrate that `DuRM` facilitates achieving a larger gradient variance during training. Further, with such a gradient distribution, our theory shows that models are inclined to converge towards better flat local minima, which has been well studied (Foret et al., 2021; Izmailov et al., 2018; Cha et al., 2021) as a more generalized model state. Empirically, we first conduct extensive experiments to validate the efficacy of `DuRM`, where the tasks include conventional classification, semantic segmentation, OOD generalization, adversarial robustness and long-tailed recognition scenarios, indicating its effectiveness in enhancing generalization in diverse scenarios. We then analyze the impact of dummy class numbers, original class numbers, training data size, and backbones. Besides, we also empirically show that `DuRM` aids the model to converge towards more flat local minima than ERM and `DuRM` remains compatible with existing generalization techniques. Note that we do not exploit any advanced training or new baselines throught the experiments, thus `DuRM` does not exhibit statet-of-the-art performance. The goal of this paper is not about achieving SOTA on these tasks, but fostering new interest in the fundamental risk minimization research.

**Contributions.** Our contributions are three-fold. 1) We devise `DuRM`, a frustratingly easy and effective technique for model generalization that is almost free lunch. 2) We present a detailed theoretical analysis of `DuRM` to guarantee its effectiveness. 3) We conduct extensive experiments to validate its performance in five diverse classification scenarios with comprehensive ablation studies.

## 2 DUMMY RISK MINIMIZATION

### 2.1 PROBLEM FORMULATION

In standard supervised classification, we are given a labelled dataset $\mathcal{D} = \{\mathbf{x}_i, y_i\}_{i=1}^N$, in which $\mathbf{x} \in \mathbb{R}^d$ is the $d$-dimensional input and $y \in \{1, \ldots, C\}$ is the output, where $C$ denotes the number of classes. To learn the map between input and output, Empirical risk minimization (ERM) (Sain, 1996) is generally employed to achieve minimal risk on $\mathcal{D}$ by learning a classifier $h_{erm} : \mathbf{x} \mapsto \mathbb{R}^C$.

Specifically, $h$ maps the original inputs to *logits* $\mathbf{z} \in \mathbb{R}^C$ that can be further transformed into classification probability, in which each element indicates the model confidence of the corresponding category. The learning objective of ERM is formulated as

$$h^* = \underset{h_{erm} \in \mathcal{H}}{\arg\min} \frac{1}{N} \sum_{i=1}^{N} \ell(h_{erm}(\mathbf{x}_i); \mathbf{y}_i), \tag{1}$$

where $\ell(\cdot, \cdot)$ is the classification loss such as cross entropy and $\mathbf{y}_i \in \mathbb{N}^C$ is the one-hot vector.

**Definition 1** (Dummy risk minimization (DuRM)). *DuRM is a frustratingly easy extension of ERM to improve its generalization ability. The core of DuRM is the newly added $C_d$ dummy classes on the original $C$ classes, i.e., DuRM solves a $C$-class classification problem using a $(C + C_d)$-class classification instantiation:*

$$h^* = \underset{\substack{h_{erm} \in \mathcal{H}, \\ h_{durm} \in \mathcal{H}}}{\arg\min} \frac{1}{N} \sum_{i=1}^{N} \left\{ \ell[cat(h_{erm}(\mathbf{x}_i), h_{erm}(\mathbf{x}_i)), cat(\mathbf{y}_i, \mathbf{0})] \right\}, \tag{2}$$

*where $h_{durm}(\mathbf{x}_i) \in \mathbb{R}^{C_d}$ is the output logits of DuRM and $\mathbf{0}$ denotes the label vector with the same dimension as $h_{durm}(\mathbf{x}_i)$, indicating that there is no supervision information for dummy classes.*

Figure 1 illustrates the main idea of DuRM. DuRM degenerates to the original ERM when $C_d = 0$.

## 2.2 DuRM AS GRADIENT REGULARIZATION

In this section, we theoretically analyze the impact of DuRM on the gradient before analyzing its contribution to generalization ability.

**No samples are classified into dummy classes in DuRM**    First, we show that *no* samples can be classified as dummy classes. Inspired by gradient decoupling in loss computing (Yang et al., 2022; Yeh et al., 2022), the gradient for one neuron in softmax with class $k \in [1, C + C_d]$ after training for one epoch is formulated as

$$-g_k = \sum_N \frac{\partial \ell}{\partial \mathbf{z}_k} = \overbrace{\sum_{i=1, c_i \neq k}^{N_{\bar{k}}} \mathbf{p}_k^{(i)}}^{\text{push term}} + \overbrace{\sum_{j=1, c_j = k}^{N_k} (\mathbf{p}_k^{(j)} - 1)}^{\text{pull term}} = \mathcal{F}_{\text{push}}(\mathbf{z}_k) + \mathcal{F}_{\text{pull}}(\mathbf{z}_k), \tag{3}$$

where $\mathbf{z}$ denotes logit and $\mathbf{p}$ is the probability vector, with subscripts $c$ and $k$ as their indices. $N_k$ and $N_{\bar{k}}$ are the number of samples belonging to and not belonging to class $k$, respectively.

Note that Eq. (3) are divided into two terms: the *push* term $\mathcal{F}_{\text{push}}(\mathbf{z}_k)$ and the *pull* term $\mathcal{F}_{\text{pull}}(\mathbf{z}_k)$. Since the gradient is taken w.r.t. the logits of class $k$, the pull term $\mathcal{F}_{\text{pull}}(\mathbf{z}_k)$ will pull the current samples toward class $k$, while the push term $\mathcal{F}_{\text{push}}(\mathbf{z}_k)$ is pushing samples away from class $k$. Because no sample should be classified into dummy classes, no $\mathcal{F}_{\text{pull}}$ will be implemented to pull samples towards the dummy class, while each sample is being pushed away from the dummy class. Therefore, no samples are classified as dummy classes in DuRM. By now, the influence of the dummy class is only on the push term $\mathcal{F}_{\text{push}}(\mathbf{z}_c)$ in loss computing.

However, the push term in Eq. (3) is difficult to quantize due to the unknown status of the deep model prediction. To better analyze the influence of DuRM on model generalization, we turn to modelling the influence on the gradient.

**DuRM aids to derive gradients with greater variance**    Without loss of generality, we assume that during training, the probability distribution of prediction confidence to class $c$ is composed of two Gaussian distributions of samples belonging and not belonging to class $c$, formulated as

$$P(\mathbf{p}_c) = \alpha \cdot \mathcal{N}(\mu_{c_n}, \sigma_{c_n}^2) + (1 - \alpha) \cdot \mathcal{N}(\mu_{c_p}, \sigma_{c_p}^2), \text{ where } \lim_{N_{c_n} \to \infty} \mu_{c_n} = 0^+, \lim_{N_{c_p} \to \infty} \mu_{c_p} = 1^-, \tag{4}$$

where $\alpha$ is the coefficient, $\mu$ and $\sigma^2$ are the mean and variance of Gaussian distribution, respectively.

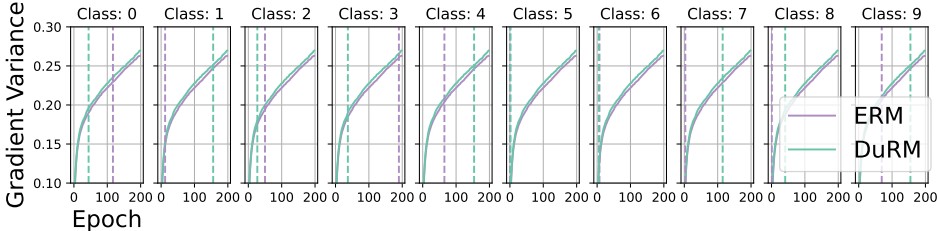

Figure 2: The convergence curve of training a ResNet-18 on CIFAR-10 with ERM and `DuRM`. Two methods are trained by 200 epochs, where we record the gradient of cross-entropy loss to logits of each category in an epoch. The dashed lines denote the epoch where the best-validated model emerged. Moreover, there are mean accuracy of $84.64\%$ and $84.76\%$ for ERM with DuRM, respectively. What has not been exhibited is that the class-wise accuracy marked in dashed lines for `DuRM` outperforms ERM, which further shows the superiority of `DuRM`. Best viewed in colour and zoomed.

Subscripts $c_{p(ositive)}$ and $c_{n(egative)}$ indicate whether samples belong to class $c$ or not. As $N \to \infty$, the mean probability of negative and positive samples are approaching to 0 and 1, respectively.

Denote $g_c$ as the gradient trained with ERM for class $c$. Then, combining Eq. (4) with Eq. (3), $g_c$ has the following probability distribution:

$$P(g_c) = \alpha \cdot \mathcal{N}(-\mu_{c_n}, \sigma^2_{c_n}) + (1 - \alpha) \cdot \mathcal{N}(1 - \mu_{c_p}, \sigma^2_{c_p}). \tag{5}$$

Recall that Eq. (3) shows the model gradient will be influenced, so we model this influence as $g_d$ and derive an `DuRM` version of the gradient as $\widehat{g}_c \triangleq g_c + g_d$. Intuitively, since there is no evidence showing that `DuRM` can increase or decrease the ERM gradient $g_c$, we can moderately assume $g_d \sim \mathcal{N}(0, \sigma^2_d)$. Now, we have the following theorem:

**Theorem 1** (`DuRM`'s influence on gradient). *Denote $g_c$ and $\hat{g}_c$ as the gradient of ERM and `DuRM` on class c, respectively. $\mathbb{E}$ and $\mathbb{D}$ are the expectation and variance, respectively. Then, the equality of $\mathbb{E}(\widehat{g}_c) = \mathbb{E}(g_c)$ and inequality of $\mathbb{D}(\widehat{g}_c) \geq \mathbb{D}(g_c)$ hold.*

*Proof.* (informal; formal proof is in Appendix A.1) Let us divide $\mathbb{D}(\widehat{g}_c)$ into three terms: $\mathbb{D}(g_c)$, $\mathbb{D}(g_d)$ and a covariance item. Since $\mathbb{D}(g_c)$ is composed of two independent sub-Gaussian distributions with zero mean, the covariance between two sub-Gaussian with $g_d$ is derived as 0. Thus, we have $\mathbb{D}(\widehat{g}_c) - \mathbb{D}(g_c) = \mathbb{D}(g_d) \geq 0$. And the expectation equality holds since $\mathbb{E}(g_d) = 0$. □

**Empirical observation on the prediction variance** We implement `DuRM` under a toy setting to analyze the sample variance. As depicted in Fig. 2, a `DuRM` solution is deployed to compare with ERM. Starting with the curve content, `DuRM` achieves a greater gradient variance for each category. Moreover, the dashed line marked the epoch when the best-validated model emerges, and `DuRM` outperforms ERM on accuracy for all categories, which is not exhibited in the figure. The toy experiment provides an empirical guarantee to the above derived theoretical results and aids us to further exploit how `DuRM` works. To control the results, we deploy a deliberate failure case of `DuRM` in Appendix C.6 for a more comprehensive analysis.

## 2.3 GENERALIZATION ANALYSIS OF `DuRM`

The previous section shows that `DuRM` derives greater variance on gradients during training. In this section, we analyze the generalization of `DuRM` by showing that the greater gradient variance brought by `DuRM` facilitates model convergence to local minima.

**Greater gradient variances facilitate convergence to flat local minima** To measure the flatness of a local minima, we first apply a two-order Taylor expansion to the loss function when gradient descent is on the local minima:

$$\ell(\mathbf{w} + \mathbf{v}) - \ell(\mathbf{w}) = \mathbf{v}^\mathsf{T}\mathbf{J}(\mathbf{w}) + \frac{1}{2}\mathbf{v}^\mathsf{T}\mathbf{H}(\mathbf{w})\mathbf{v} + o(\|\mathbf{v}\|^2), \tag{6}$$

where $\mathbf{w}$ is the model weights on the local minima, $\mathbf{v}$ is a weight step implemented upon $\mathbf{w}$ to escape the local minima. $\mathbf{J}(\mathbf{w})$ and $\mathbf{H}(\mathbf{w})$ are Jacobian and Hessian matrix, respectively. Then, the

convergence state of a model can be recognized as the stability under the sense of Lyapunov (Shevitz & Paden, 1994):

**Definition 2** (Stability under the Sense of Lyapunov (Shevitz & Paden, 1994)). *The equilibrium point* **w** *at the original time is stable under the sense of Lyapunov, i.i.f.:*

$$\forall \mathbf{v}, \text{s. t. } \|\mathbf{v}\| \leq \delta(\mathbf{v}) \Rightarrow \|\ell(\mathbf{w} + \mathbf{v}) - \ell(\mathbf{w})\| \leq \varepsilon, \tag{7}$$

*where $\delta(\cdot)$ limits the upper bound of disturbance to* **v** *as a small scale and $\varepsilon$ is defined as the* stability.

In `DuRM`, when **w** is the local minima, the model can then escape the local minima with a step disturbance of **v**:

$$\varepsilon \leq \ell(\mathbf{w} + \mathbf{v}) - \ell(\mathbf{w}), \text{s. t. } \ell(\mathbf{w} + \mathbf{v}) > \ell(\mathbf{w}). \tag{8}$$

To this end, we are able to give the definition of flatness.

**Definition 3** (Flatness of Loss Landscape in Local Minima). *Given a model parameter* **w** *in flat local minima, then, manually deploying a weight disturbance* **v** *to* **w** *makes* **w** + **v** *escape the local minima and meet Eq. equation 8. The flatness of loss landscape in given local minima $\tau \propto \frac{1}{\varepsilon}$.*

Recall Eq. equation 8, since **w** is on the local minima, we have $\det(\mathbf{J}) = 0$. To better understand how `DuRM` influences the flatness of local minima $\varepsilon$, we need to scale up the inequality. Then, suppose $\rho$ is the greatest eigenvalue to Hessian matrix. We can loosen the upper bound of $\varepsilon$ as:

$$\varepsilon \leq \frac{\rho \|\mathbf{v}\|^2}{2} + o(\|\mathbf{v}\|^2) \approx \frac{\rho \|\mathbf{v}\|^2}{2}. \tag{9}$$

Hence, to achieve a more flat (greater $\tau$) local minima, there should be a smaller $\varepsilon$. Furthermore, achieving a tighter upper bound of $\varepsilon$ could be such a solution. Recall Thm. 1 that shows `DuRM` works as a gradient regularization by improving the gradient variance. With respect to $\rho$, its corresponding eigenvector denotes the steepest direction, which is also the gradient direction. Therefore, we clarify the correlation between g with $\rho$ in the following proposition, whose proof is in Appendix A.2.

**Proposition 1.** *Let* $\mathbf{H}(\mathbf{w})$ *be the Hessian matrix for the model with parameter* **w**. *With the definition of eigenvector* **s** *and greatest eigenvalue $\rho$ of* $\mathbf{H}(\mathbf{w})\mathbf{s} = \rho\mathbf{s}$, *there lies a positive correlation of $\rho \propto g$.*

When the model converges to a local minimum, we can assume the step has the minimum gradient $g_{(1)}$ in $T$ steps. To achieve a flat local minimum, there should be a tighter upper bound of $\varepsilon$ to achieve a better flatness $\tau$. Then, according to Prop. 2 and the above assumption, a smaller $g_{(1)}$ could be the solution. To parameterize this issue, we assume $g \sim \mathcal{N}(\mu_g, \sigma_g^2)$[1]. Hence, comparing gradients $g$ for ERM and $\widehat{g}$ for our `DuRM`, the population mean along with variance have $\mu = \widehat{\mu}$ and $\sigma^2 < \widehat{\sigma}^2$ respectively. Given a fixed model initialization point, let ERM and `DuRM` optimize such a model with equal steps $T$ and we can sample $T$ gradient points. We have the minimal order statistics among $T$ samples as $g_{(1)}$ and $\widehat{g}_{(1)}$, whose probability density function can be derived as:

$$f_{g_{(1)}}(g) = T \left[1 - F_g(g)\right]^{T-1} \cdot f_g(g)$$
$$= T[1 - \frac{1}{\sqrt{2\pi\sigma^2}} \int_{-\infty}^{g} \exp(-\frac{(g-\mu)^2}{2\sigma^2}) \mathrm{d}g]^{T-1} \cdot \frac{1}{\sqrt{2\pi\sigma^2}} \cdot \exp(-\frac{(g-\mu)^2}{2\sigma^2}), \tag{10}$$

where $F$ is the distribution function. Similar formulation goes to $f_{\widehat{g}_{(1)}}(g)$ by replacing $\mu, \sigma$ with $\widehat{\mu}, \widehat{\sigma}$. Then, we derive the following theorem to show how that `DuRM` obtains a smaller $g_{(1)}$.

**Theorem 2** (`DuRM` obtains better local minima). *Given gradients $g \sim \mathcal{N}(\mu_1, \sigma_1^2)$ and $\widehat{g} \sim \mathcal{N}(\mu_2, \sigma_2^2)$, where $\mu_1 = \mu_2$ and $\sigma_1 < \sigma_2$. Assume gradient descent has $T$ steps. The inequality between minimum order statistics of empirical gradients holds with a high probability of $P(g_{(1)} \geq \widehat{g}_{(1)}) = \frac{1}{2} + T \int [1 - F_g(\widehat{g})]^T - [1 - F_{\widehat{g}}(\widehat{g})]^T \cdot (1 - F_{\widehat{g}_{(1)}})^{T-1} \mathrm{d}F_{\widehat{g}_{(1)}}$, such a probability owns a numerical solution of $0.5 + \Theta(\frac{\sigma_1}{\sigma_2})$, where $\Theta(\frac{\sigma_1}{\sigma_2}) \geq 0, \forall \sigma_1 < \sigma_2 \in \mathbb{R}$.*

Proof can be found in Appendix A.3. Then, considering the positive correlation between $\rho$ with $g_{(1)}$, the model with greater gradient variance is inclined to derive a smaller $\rho$, in which the $\varepsilon$ is with a tighter upper bound and a better flatness $\tau$ is achieved. Then, (Foret et al., 2021; Izmailov et al., 2018; Cha et al., 2021) have shown theoretical or empirical result in demonstrating that a more flat local minima is a more generalized model state.

---

[1]Since Eq. equation 5 demonstrates that the mean values of two sub-Gaussian distribution are aligned with each other, a mixed Gaussian distributions can be approximately degenerated to a Gaussian distribution.

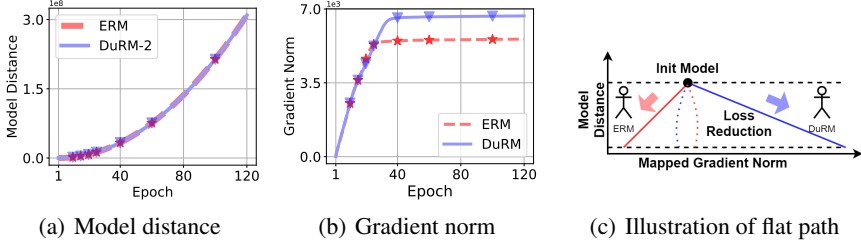

(a) Model distance      (b) Gradient norm      (c) Illustration of flat path

Figure 3: (a) The curve change of model distance during training. (b) The curve change of cumulative gradient norm along training epoch. (c) An illustration on convergence with a more flat path.

Table 1: Accuracy on different classification datasets.

| Dataset | Model | ERM | DuRM | Dataset | Model | ERM | DuRM |
|---|---|---|---|---|---|---|---|
| ImageNet-1K | ResNet18 | $69.74_{\pm0.09}$ | $\mathbf{69.76}_{\pm0.06}$ | Oxford-Pet | ResNet-18 | $90.23_{\pm0.19}$ | $\mathbf{90.30}_{\pm0.21}$ |
| CIFAR-10 | ResNet-18 | $82.08_{\pm0.34}$ | $\mathbf{82.11}_{\pm0.30}$ | Oxford-Pet | ResNext-50 | $93.51_{\pm0.16}$ | $\mathbf{93.57}_{\pm0.17}$ |
| CIFAR-10 | ViT-Tiny | $92.79_{\pm0.67}$ | $\mathbf{94.38}_{\pm0.44}$ | Oxford-Pet | MLP-Mixer-B | $\mathbf{85.06}_{\pm0.33}$ | $84.71_{\pm0.47}$ |
| CIFAR-10 | ResNet-50 | $\mathbf{84.24}_{\pm0.22}$ | $84.10_{\pm0.57}$ | Flower-102 | ResNet-18 | $85.07_{\pm0.24}$ | $\mathbf{85.08}_{\pm0.19}$ |
| CIFAR-100 | ResNet-18 | $58.42_{\pm0.33}$ | $\mathbf{58.50}_{\pm0.38}$ | Flower-102 | ResNet-101 | $86.69_{\pm0.23}$ | $\mathbf{86.91}_{\pm0.12}$ |
| CIFAR-100 | ResNet-50 | $60.17_{\pm2.06}$ | $\mathbf{61.03}_{\pm0.11}$ | Flower-102 | ViT-Tiny | $87.33_{\pm1.56}$ | $\mathbf{88.71}_{\pm1.17}$ |
| CIFAR-100 | Resnext-50 | $\mathbf{65.02}_{\pm0.35}$ | $64.44_{\pm0.10}$ | THUCNews | Transformer | $89.55_{\pm0.46}$ | $\mathbf{89.92}_{\pm0.31}$ |
| STL-10 | Swin-Tiny | $95.55_{\pm0.47}$ | $\mathbf{96.33}_{\pm0.14}$ | UrbanSound8K | LSTM | $65.10_{\pm0.08}$ | $\mathbf{66.00}_{\pm1.01}$ |
| STL-10 | MobileViT-S | $94.32_{\pm0.15}$ | $\mathbf{95.03}_{\pm0.53}$ | STL-10 | ResNet-18 | $85.20_{\pm0.75}$ | $\mathbf{86.21}_{\pm0.73}$ |

**Empirical evidence to flat local minima** Inspired by (Liu et al., 2023), we empirically show that dummy class could converge to a flat local minima. First, we measure the gradient norm during training, denoted as $\|\mathbf{w}\|^2$. Then, we compute the model distance between time step $t$ and the initialized weight as $\|\mathbf{w}_t - \mathbf{w}_0\|^2$. As shown in Figure 3, the model distances vary in the same path between ERM and DuRM. However, DuRM has a greater gradient norm than ERM, indicating that DuRM has more flatness.

## 3 EXPERIMENTS

In this section, we conduct extensive experiments to evaluate DuRM on diverse tasks: conventional sample-wise classification, semantic segmentation, out-of-distribution generalization, adversarial robustness, and long-tailed recognition. Then, we present thorough synopsis to explore why and how DuRM works. All experiments are repeated three times with different seeds to control randomness.

### 3.1 MAIN RESULTS

**Conventional Sample-wise Classification** First of all, we apply DuRM to conventional (sample-wise) classification tasks. We adopt CIFAR-10/100 (Krizhevsky et al., 2009), STL-10 (Coates et al., 2011), Oxford-Pet-35 (Parkhi et al., 2012), Flower-102 (Nilsback & Zisserman, 2008), ImageNet-1K (Deng et al., 2009), THUCNews-10 (Li & Sun, 2007), and UrbanSound8K-10 (Salamon & Bello, 2017) datasets. Different backbones are used for different datasets spanning from ResNet (He et al., 2016), ResNext (Xie et al., 2017) to Transformer (Vaswani et al., 2017), LSTM (Shi et al., 2015) (ViT (Dosovitskiy et al., 2021)), Swin (Liu et al., 2021), Mobile ViT (Mehta & Rastegari, 2022) and MLP-Mixer (Tolstikhin et al., 2021), providing a holistic evaluation across datasets and backbones.

Table 1 shows the Top-1 classification accuracy using DuRM-2 (i.e., adding 2 dummy classes) without adding specific generalization techniques such as EMA and mixup (Zhang et al., 2017). Experiments in Table 7 show that DuRM can also work seamlessly with these techniques. Results of using other dummy classes are presented in Appendix C.1. It is noted that DuRM outperforms ERM in most cases (39 out of 52, combining Appendix C.1) and it reduces the variance of the results.

**Semantic Segmentation** Then, we apply DuRM to semantic segmentation tasks. We leverage three popular benchmarks: CityScapes (Cordts et al., 2016), Pascal-VOC 2012 (Everingham et al., 2015), and LoveDA (Wang et al., 2021). DuRM is plugged into different segmentation algorithms: FCN-ResNet101 (Long et al., 2015), Segformer B1 and B5 (Xie et al., 2021), Deeplab-V3 with a ResNet101 (Chen et al., 2017), HRNet-W48 (Wang et al., 2020) and PSPNet coordinating with a

Table 3: Results on CityScapes and Pascal VOC semantic segmentation datasets.

| CityScapes | mIoU (%) | | mACC (%) | | Pascal VOC | mIoU (%) | | mACC (%) | |
|---|---|---|---|---|---|---|---|---|---|
| | Val | Test | Val | Test | | Val | Test | Val | Test |
| FCN-R101 | 64.05 | 61.35 | 70.97 | - | DLB-R101 | **63.72** | 61.55 | **74.17** | - |
| +DuRM | **65.05** | **63.35** | **72.36** | - | +DuRM | 63.54 | **61.64** | 73.29 | - |
| Mit B0 | 73.57 | 73.56 | 81.88 | - | HRNet-W48 | 57.53 | 57.78 | 65.24 | - |
| +DuRM | **75.99** | **74.23** | **83.92** | - | +DuRM | **59.16** | **59.63** | **66.62** | - |
| MiT B5 | 81.82 | 80.66 | 88.69 | - | PSPNet-R18 | 57.87 | 56.29 | 68.83 | - |
| +DuRM | **82.08** | **80.88** | **88.90** | - | +DuRM | **57.98** | **56.96** | **68.92** | - |
| AVG improvement | 1.23 | 0.96 | 1.21 | - | AVG improvement | 0.52 | 0.87 | 0.19 | - |

Table 4: Results on out-of-distribution generalization datasets.

| Dataset | Method | Vanilla | SWAD | DANN | VReX | RSC | MMD |
|---|---|---|---|---|---|---|---|
| VLCS | ERM | $\textbf{72.42}_{\pm0.09}$ | $75.79_{\pm0.57}$ | $69.29_{\pm0.96}$ | $\textbf{75.16}_{\pm0.67}$ | $74.87_{\pm0.56}$ | $74.39_{\pm0.74}$ |
| | +DuRM | $71.94_{\pm0.85\downarrow}$ | $\textbf{76.50}_{\pm0.23\uparrow}$ | $\textbf{69.59}_{\pm1.36\uparrow}$ | $74.92_{\pm0.87\downarrow}$ | $\textbf{75.20}_{\pm0.31\uparrow}$ | $\textbf{75.15}_{\pm0.86\uparrow}$ |
| OfficeHome | ERM | $62.65_{\pm0.21}$ | $61.86_{\pm0.15}$ | $59.38_{\pm0.38}$ | $63.27_{\pm0.21}$ | $61.30_{\pm0.10}$ | $\textbf{63.46}_{\pm0.15}$ |
| | +DuRM | $\textbf{62.93}_{\pm0.23\uparrow}$ | $\textbf{62.07}_{\pm0.11\uparrow}$ | $\textbf{59.70}_{\pm0.15\uparrow}$ | $\textbf{63.46}_{\pm0.24\uparrow}$ | $\textbf{61.60}_{\pm0.28\uparrow}$ | $63.12_{\pm0.11\downarrow}$ |
| PACS | ERM | $81.78_{\pm0.21}$ | $82.96_{\pm0.32}$ | $77.98_{\pm0.96}$ | $\textbf{81.13}_{\pm0.78}$ | $\textbf{80.97}_{\pm0.91}$ | $81.10_{\pm0.43}$ |
| | +DuRM | $\textbf{81.86}_{\pm0.54\uparrow}$ | $\textbf{83.22}_{\pm0.28\uparrow}$ | $\textbf{78.58}_{\pm1.30\uparrow}$ | $80.81_{\pm0.56\downarrow}$ | $80.29_{\pm0.91\downarrow}$ | $\textbf{81.47}_{\pm0.16\uparrow}$ |
| TerraInc | ERM | $\textbf{41.66}_{\pm1.39}$ | $41.84_{\pm0.58}$ | $33.11_{\pm2.39}$ | $42.00_{\pm0.83}$ | $44.44_{\pm1.41}$ | $37.47_{\pm1.28}$ |
| | +DuRM | $40.71_{\pm1.26\downarrow}$ | $\textbf{42.88}_{\pm0.77\uparrow}$ | $32.36_{\pm3.39\downarrow}$ | $\textbf{42.19}_{\pm1.56\uparrow}$ | $\textbf{44.77}_{\pm1.35\uparrow}$ | $\textbf{37.67}_{\pm1.45\uparrow}$ |

backbone model of ResNet18 (Zhao et al., 2017), UperNetSwin (Wang et al., 2023), and Segmentor (Strudel et al., 2021). We borrow the codebase from MMSegmentaion (Contributors, 2020) for implementation.[2]

Table 3 shows the mIoU and mACC on CityScapes and Pascal VOC datasets using `DuRM`-1. Detailed class-wise results are presented in Appendix C.2. We see that `DuRM` can boost the performance of most segmentation algorithms. Moreover, LoveDA (Wang et al., 2021) is more challenging since it is composed of remote sensing images which are more complex. It further contains a special class *Background* filled with unannotated objects. As shown in Table 2, `DuRM` achieves the largest

Table 2: Results on LoveDA dataset.

| Method | UperNetSwin | +DuRM | Segmentor | +DuRM |
|---|---|---|---|---|
| Background | 44.75 | **45.65** | 43.93 | **45.27** |
| Building | **57.47** | 55.85 | **59.03** | 58.79 |
| Road | **56.40** | 56.22 | 57.00 | **57.77** |
| Water | **78.39** | 77.24 | 79.01 | **80.28** |
| Barren | 12.39 | **16.65** | 16.69 | **18.36** |
| Forest | 45.31 | **46.43** | 45.76 | **45.85** |
| Agriculture | 56.52 | **56.71** | 60.94 | **61.04** |
| mIOU | 50.18 | **50.68** | 51.76 | **52.48** |

improvement on *Background* and *Barren* (a hard category), when coordinating with (Wang et al., 2023) and (Strudel et al., 2021). The surprise is, when implementing Segmentor + `DuRM`-1 on LoveDA, it outputs samples in the dummy class to some unannotated pixels, indicating that that `DuRM` can facilitate the model to learn representations, serving beyond as a fine-grained classifier. We further show visualizations in Appendix C.2.1 to better support our claim.

**Out-of-distribution generalization** We then apply `DuRM` to OOD generalization that evaluates generalization under distribution shifts. Following previous works (Gulrajani & Lopez-Paz, 2020), we use ResNet-18 as the backbone and then plug `DuRM` in some state-of-the-art OOD approaches including ERM, DANN (Ajakan et al., 2016), VRex (Krueger et al., 2021), RSC (Huang et al., 2020), MMD (Tzeng et al., 2014), and SWAD (Cha et al., 2021). The evaluation datasets include VLCS (Fang et al., 2013), OfficeHome (Venkateswara et al., 2017), PACS (Li et al., 2017), and TerraInc (Beery et al., 2018).

Table 4 shows the averaged results on each datasets using `DuRM`-1 and the details results on each domain are in Appendix C.3. We observe that `DuRM` can boost the performance of existing algorithms in OOD generalization tasks in diverse distribution shift settings ranging from natural shift (e.g., Office-Home) to correlation shift (e.g., TerraInc). This again proves its effectiveness.

---

[2]All datasets should be tested online limited submission trials, we did not use multi-trail but fixed the seed to promise reproduction. Moreover, the '-' in Table 3 are because the online tested results did not feed them back.

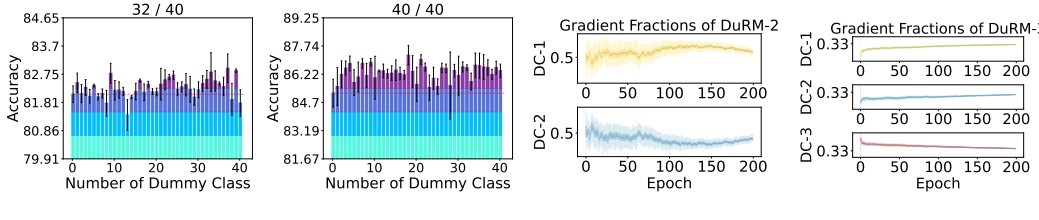

(a) CIFAR10 DuRM Swap    (b) STL DuRM Swap    (c) DuRM-2 Grad. Fraction (d) DuRM-3 Grad. Fraction

Figure 4: (a) and (b) are accuracy varying curve along the number of dummy class on CIFAR10 and STL, in which the titles denote the counts of DuRM outperforming ERM. The different colors are to distinguish the accuracy range. (c) and (d) gradient fraction convergence curves for dummy classes under DuRM-2 and DuRM-3, respectively.

**Adversarial robustness** We further evaluate DuRM for adversarial robustness (Madry et al., 2017), where two popular attacks, FGSM (Goodfellow et al., 2015) and PGD (Carlini & Wagner, 2017) are adopted. We conduct experiments on CIFAR-10 (Krizhevsky et al., 2009) following previous works (Croce et al., 2021). We perform DuRM in both attack and defense (i.e., use adversarial training on FGSM and PGD) scenarios and report the robust accuracy. Results in Table 5 demonstrate that DuRM is more robust to these attacks, showing its efficacy in handling adversarial perturbations.

Table 5: Results of adversarial robustness (CIFAR-10).

| FGSM Attack | | FGSM + AT | |
|---|---|---|---|
| FGSM | +DuRM | FGSM | +DuRM |
| $28.49_{\pm0.25}$ | $\mathbf{30.13}_{\pm0.52}$ | $53.54_{\pm0.25}$ | $\mathbf{54.18}_{\pm0.31}$ |
| PGD Attack | | PGD + AT | |
| PGD | +DuRM | PGD | +DuRM |
| $12.55_{\pm0.47}$ | $\mathbf{13.79}_{\pm0.67}$ | $46.85_{\pm0.19}$ | $\mathbf{47.10}_{\pm0.38}$ |

**Long-tailed recognition** Finally, we evaluate DuRM in long-tailed recognition tasks. We implement three different imbalanced ratios[3] on CIFAR-10-LT (Krizhevsky et al., 2009), following previous works (Yang & Xu, 2020; Tang et al., 2020). As shown in Table 6, DuRM is consistently more robust than ERM in different imbalance settings.

Table 6: Results on long-tailed CIFAR-10 dataset.

| Imb. ratio | vanilla | +DuRM |
|---|---|---|
| 100 | $63.29_{\pm0.82}$ | $\mathbf{63.33}_{\pm0.33}$ |
| 50 | $69.63_{\pm0.21}$ | $\mathbf{69.97}_{\pm0.75}$ |
| 10 | $81.01_{\pm0.34}$ | $\mathbf{81.15}_{\pm0.46}$ |

### 3.2 How Many Dummy Classes Do You Need?

Now we discuss the influence of the number of dummy classes $C_d$. To exploit an empirical answer, we run extensive experiments by varying $C_d$ from 1 to 40 on three datasets, namely CIFAR-10 in Figure 4(a), STL in 4(b) and Oxford-Pet in Appendix C.7. As shown, it reveals that there is *no* explicit correlation between the number of dummy classes and model performance. To this end, we assert that the number of dummy classes is not a latent factor influencing the model generalization. **Therefore, DuRM can be easily used in real applications without extensively setting the number of dummy classes.** Additionally, the ablation results in Figure 5 further support this empirical conclusion.

### 3.3 Analyzing the Gradient Fraction on Dummy Classes

Based on the above conclusion, there may arise a question of whether only head dummy classes are effective in DuRM, resulting in there goes no explicit correlation between model generalization with dummy class quantity. To answer that question, we record the gradient fraction for multiple dummy class settings, i.e., the gradient contribution of each dummy class is recorded with $C_d \in \{2, 3, 4\}$, as shown in Figure 4(c) and 4(d). The gradient fraction for each dummy class can be derived via the fraction between prediction probability of corresponding category with the sum probability to all dummy classes. Please note that the results of $C_d = 4$ is in Appendix C.8. Results show that the fraction is converging towards the *averaging* level, which means that all dummy classes are treated *equally*. Hence, each dummy class in DuRM is equally effective. As for the results variance among different dummy classes, this could be understood from the perspective of gradient transport randomness, which is reflected as the convergence curve of gradient fraction in Figure 4(c) and 4(d).

---

[3]Imbalance ratio is the ratio between the number of samples of the head (biggest) class with tail (smallest) class, e.g., there are 1000 samples for head and 10 samples for tail, resulting an imbalanced ratio of 100.

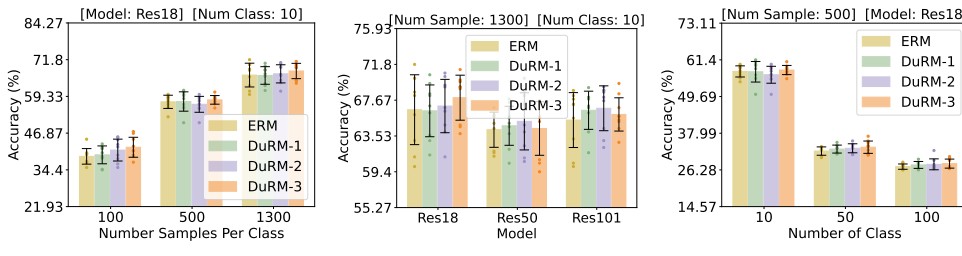

(a) Control Number of Samples    (b) Control Model Scales    (c) Control Number of Class

Figure 5: The controlled experiments on manually generated subsets from ImageNet. Under each data-centric setting, we generate three different datasets. Then, we also repeat each experiments with three trials. To this end, the bars are averaged under nine values.

### 3.4 Ablation Study on Model Scale, Number of Class, and Data Scale

To better understand `DuRM` and provide a usage guideline, we conduct comprehensive ablation experiments to perform controlled analysis on factors including model scale, the number of original class and dataset scale. We manually generate some subsets from ImageNet-1K (Deng et al., 2009). For each factor, we pick three levels of options. Moreover, to eliminate bias from the generated subset, we generate three different subsets under each data-related factor and repeat the experiments with three trials.

As shown in Figure 5(a), we control the number of samples per class. Comparing the improvement under 100 with 1300, we notice that `DuRM` achieves larger improvement under smaller data scale. Then, the results of model scale in Figure 5(b) reflect that `DuRM` works better under more over-fitting scenarios. Later, we control the number of class in Figure 5(c), where the results further reveals better performance of `DuRM` under over-fitting. Therefore, summarizing all the results, it is demonstrated that our `DuRM` is effective under multi-scene on data-centric or model-centric variance. To this end, the above results facilitates us demonstrating that `DuRM` works as a model regularization, especially in reducing over-fitting to enhance generalization.

### 3.5 Compatibility with Other Regularization Technique

In this subsection, we plug our `DuRM` into some popular regularization technique to verify the compatibility of `DuRM`. Concretely, we plug `DuRM` into: Early Stop, L2 Regularization, Gradient Momentum, Mixup (Zhang et al., 2017), Exponentially Momentum Average (EMA), Stochastic Weight Average (SWA) (Izmailov et al., 2018). We conduct experiments on CIFAR-10 dataset with ERM as the baseline. As shown in Figure 7, `DuRM` can be seamlessly integrated into existing regularization technique to achieve further improvements. This indicates that `DuRM` can be easily applied to a broad range of applications for better performance.

Table 7: Compatibility to other regularization on CIFAR-10.

| Method | ERM | +DuRM |
|---|---|---|
| Vanilla | $79.63_{\pm 0.09}$ | $\mathbf{80.01}_{\pm 0.06}$ |
| EarlyStop | $78.68_{\pm 0.35}$ | $\mathbf{78.96}_{\pm 0.12}$ |
| L2 | $79.70_{\pm 0.25}$ | $\mathbf{79.78}_{\pm 0.21}$ |
| Momentum | $81.98_{\pm 0.18}$ | $\mathbf{82.33}_{\pm 0.30}$ |
| MixUp | $\mathbf{82.91}_{\pm 0.86}$ | $82.76_{\pm 0.81}$ |
| EMA | $79.62_{\pm 0.17}$ | $\mathbf{79.92}_{\pm 0.38}$ |
| SWA | $93.46_{\pm 0.17}$ | $\mathbf{93.48}_{\pm 0.14}$ |

## 4 Conclusion

This paper is motivated by the unsatisfactory generalization capacity of ERM in classification tasks. We devised a frustratingly easy method called Dummy Risk Minimization (`DuRM`), which is almost a free lunch solution upon ERM to achieve a better generalization. `DuRM` adds several dummy classes to logits without modifying the original label space. We then provided theoretical analysis on `DuRM` in pushing model convergence towards a better flat local minimum. Empirically, we conducted extensive experiments to support our theory and provided an empirical understanding of `DuRM`. In a nutshell, this paper provided the community with an interpretable and free-lunch regularization to classification models. We hope this method could inspire new interest on generalization research.

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

# Appendix: Frustratingly Easy Model Generalization by Dummy Risk Minimization

**Overview**   This is the Appendix for Submission 4297, named "Frustratingly Easy Model Generalization by Dummy Risk Minimization". Concretely, this file is organized as following:

- Proofs to proposed theorems and propositions;
- Implementation details to experiments conducted in Sec. 3;
- Additional experimental results and analysis.

## A   PROOF

### A.1   PROOF TO THM. 3

**Theorem 3** (DuRM's influence on gradient). *Denote $g_c$ and $\hat{g}_c$ as the gradient of ERM and DuRM on class $c$, respectively. $\mathbb{E}$ and $\mathbb{D}$ are the expectation and variance, respectively. Then, the equality of $\mathbb{E}(\hat{g}_c) = \mathbb{E}(g_c)$ and inequality of $\mathbb{D}(\hat{g}_c) > \mathbb{D}(g_c)$ hold.*

*Proof.* To prove Thm. 3, let us start from understanding Eq. equation 3. Given a data point sampled $\mathbf{x}$ along with its one-hot label $\mathbf{y}$, where its scale value $\mathbf{y}_k = 1$, the classifier $h$ maps the $\mathbf{x}$ into logits $\mathbf{z}$. Then, a softmax further transfers $\mathbf{z}$ into probability $\mathbf{p}$, where each elem $\mathbf{p}_c$ is the model confidence to $\mathbf{x}$ of being class $c$. Therefore, the gradient $g$ of cross entropy loss to logits $\mathbf{z}$ can be derived as:

$$\frac{\partial \ell}{\partial \mathbf{z}} = \frac{\partial \ell}{\partial \mathbf{p}} \cdot \frac{\partial \mathbf{p}}{\partial \mathbf{z}} = -(\mathbf{p} - \mathbf{y}). \tag{11}$$

When given the whole dataset $\mathcal{D}$, the gradient for class $k$ can be summed as:

$$-g_k = \sum_N \frac{\partial \ell}{\partial z_k} = \overbrace{\sum_{i=1, c_i \neq k}^{N_{\bar{k}}} \mathbf{p}_k^{(i)}}^{\text{push term}} + \overbrace{\sum_{j=1, c_j=k}^{N_k} (\mathbf{p}_k^{(j)} - 1)}^{\text{pull term}} = \mathcal{F}_{\text{push}}(\mathbf{z}_k) + \mathcal{F}_{\text{pull}}(\mathbf{z}_k), \tag{12}$$

which is Eq. equation 3 in the main text. Then, let us put gradient Eq. equation 12 aside first. The prediction probability distribution $P(\mathbf{p}_c)$ is approximated as:

$$P(\mathbf{p}_c) = \alpha \cdot \mathcal{N}(\mu_{c_n}, \sigma_{c_n}^2) + (1 - \alpha) \cdot \mathcal{N}(\mu_{c_p}, \sigma_{c_p}^2). \tag{13}$$

Let us combine Eq. equation 13 with equation 12, then, the distribution of gradient $P(\mathbf{g}_c)$ for class $c$ can be derived as:

$$P(g_c) = \alpha \cdot \mathcal{N}(-\mu_{c_n}, \sigma_{c_n}^2) + (1 - \alpha) \cdot \mathcal{N}(1 - \mu_{c_p}, \sigma_{c_p}^2). \tag{14}$$

Thus, we model the $g_c \sim \alpha \cdot \mathcal{N}(0, \sigma_{c_n}^2) + (1 - \alpha) \cdot \mathcal{N}(0, \sigma_{c_p}^2)$, where the two sub-Gaussian are independent. Then, when adding DuRM upon ERM, there goes with a gradient regularization of $g_d \sim \mathcal{N}(0, \sigma_d^2)$. A gradient of $\hat{g}_c$ for whole DuRM gradient can be derived as

$$\begin{aligned}
\hat{g}_c = g_c + g_d &\Rightarrow \mathbb{E}(\hat{g}_c) = \mathbb{E}(g_c) \\
\mathbb{D}(\hat{g}_c) &= \mathbb{D}(g_c) + \mathbb{D}(g_d) + 2\text{Cov}(g_c, g_d) \\
&= \alpha^2 \sigma_{c_n}^2 + (1-\alpha)^2 \sigma_{c_p}^2 + \sigma_d^2 + \\
&\quad 2\alpha \left[ \mathbb{E}(g_{c_n} \cdot g_d) - \mathbb{E}(g_{c_n}) \cdot \mathbb{E}(g_d) \right] + \\
&\quad 2(1-\alpha) \left[ \mathbb{E}(g_{c_p} \cdot g_d) - \mathbb{E}(g_{c_p}) \cdot \mathbb{E}(g_d) \right],
\end{aligned} \tag{15}$$

where $\mathbb{E}(g_{c_n}), \mathbb{E}(g_{c_p}), \mathbb{E}(g_d)$ equal to zero. To further understand Eq. equation 15, we need to put our concentration on $\mathbb{E}(g_{c_n} \cdot g_d)$ and $\mathbb{E}(g_{c_p} \cdot g_d)$. To derive the item, we analyze the general form which is the product between two Gaussian distribution.

Given two Gaussian distribution, whose probability density function is as Eq. equation 16, which is also their product result.

$$
\begin{aligned}
f_p(g) \cdot f_n(g) &= \frac{1}{2\pi\sigma_p\sigma_n} \exp\left[-\left(\frac{(g-\mu_p)^2}{2\sigma_p^2} + \frac{(g-\mu_n)^2}{2\sigma_n^2}\right)\right] \\
&= \frac{1}{2\pi\sigma_p\sigma_n} \exp\left[\frac{(\sigma_n^2+\sigma_p^2)g^2 - 2(\mu_p\sigma_n^2+\mu_n\sigma_p^2)g + (\mu_p^2\sigma_p^2+\mu_n^2\sigma_p^2)}{2\sigma_n^2\sigma_p^2}\right] \\
&= \frac{1}{2\pi\sigma_p\sigma_n} \exp\left[\frac{g^2 - 2g\frac{\mu_p\sigma_n^2+\mu_n\sigma_p^2}{\sigma_n^2+\sigma_p^2} + \frac{\mu_p^2\sigma_n^2+\mu_n^2\sigma_p^2}{\sigma_n^2+\sigma_p^2}}{\frac{2\sigma_n^2\sigma_p^2}{\sigma_n^2+\sigma_p^2}}\right] \\
&= \frac{1}{2\pi\sigma_p\sigma_n} \exp\left[\frac{\left(g-\frac{\mu_n\sigma_p^2+\mu_p\sigma_n^2}{\sigma_n^2+\sigma_p^2}\right)^2 + \frac{\mu_n\sigma_p^2+\mu_p\sigma_n^2}{\sigma_n^2+\sigma_p^2} - \left(\frac{\mu_n\sigma_p^2+\mu_p\sigma_n^2}{\sigma_n^2+\sigma_p^2}\right)^2}{\frac{2\sigma_n^2\sigma_p^2}{\sigma_n^2+\sigma_p^2}}\right] \\
&= \frac{1}{2\pi\sigma_p\sigma_n} \exp\left[\frac{\left(g-\frac{\mu_n\sigma_p^2+\mu_p\sigma_n^2}{\sigma_n^2+\sigma_p^2}\right)^2}{\frac{2\sigma_n^2\sigma_p^2}{\sigma_n^2+\sigma_p^2}} + \frac{\frac{\mu_n\sigma_p^2+\mu_p\sigma_n^2}{\sigma_n^2+\sigma_p^2} - \left(\frac{\mu_n\sigma_p^2+\mu_p\sigma_n^2}{\sigma_n^2+\sigma_p^2}\right)^2}{\frac{2\sigma_n^2\sigma_p^2}{\sigma_n^2+\sigma_p^2}}\right] \\
&= \frac{1}{2\pi\sigma_p\sigma_n} \exp\left[\frac{\left(g-\frac{\mu_n\sigma_p^2+\mu_p\sigma_n^2}{\sigma_n^2+\sigma_p^2}\right)^2}{\frac{2\sigma_n^2\sigma_p^2}{\sigma_n^2+\sigma_p^2}} + \frac{(\mu_n^2\sigma_p^2+\mu_p^2\sigma_n^2)(\sigma_n^2+\sigma_p^2) - (\mu_n\sigma_p^2+\mu_p\sigma_n^2)^2}{2\sigma_n^2\sigma_p^2(\sigma_n^2+\sigma_p^2)}\right] \\
&= \frac{1}{2\pi\sigma_p\sigma_n} \exp\left[\frac{\left(g-\frac{\mu_n\sigma_p^2+\mu_p\sigma_n^2}{\sigma_n^2+\sigma_p^2}\right)^2}{\frac{2\sigma_n^2\sigma_p^2}{\sigma_n^2+\sigma_p^2}} + \frac{(\mu_p-\mu_n)^2}{2\sigma_p^2+2\sigma_n^2}\right] \\
&= \frac{1}{2\pi\sigma_p\sigma_n} \exp\left[\frac{\left(g-\frac{\mu_n\sigma_p^2+\mu_p\sigma_n^2}{\sigma_n^2+\sigma_p^2}\right)^2}{\frac{2\sigma_n^2\sigma_p^2}{\sigma_n^2+\sigma_p^2}}\right] \cdot \exp\left[\frac{(\mu_p-\mu_n)^2}{2\sigma_p^2+2\sigma_n^2}\right].
\end{aligned}
\tag{16}
$$

According to Eq. equation 16, let us substitute $g_{c_n}, g_{c_p}, g_d$ into it and derive:

$$
\begin{aligned}
\mathbb{E}(g_{c_n} \cdot g_d) &= C_n \cdot \frac{-\mu_{c_n}\sigma_d^2 + \mu_d\sigma_{c_n}^2}{\sigma_d^2 + \sigma_{c_n}^2} = 0 \\
\mathbb{E}(g_{c_p} \cdot g_d) &= C_p \cdot \frac{(1-\mu_{c_p})\sigma_d^2 + \mu_d\sigma_{c_p}^2}{\sigma_d^2 + \sigma_{c_p}^2} = 0,
\end{aligned}
\tag{17}
$$

where $C_p, C_n$ are two constants. Let us put Eq. equation 17 back to Eq. equation 15, we derive :·

$$
\mathbb{D}(\widehat{g}_c) = \alpha^2\sigma_{c_n}^2 + (1-\alpha)^2\sigma_{c_p}^2 + \sigma_d^2 \geq \mathbb{D}(g_c),
\tag{18}
$$

where the equality holds only when $\sigma_d = 0$.

$\square$

## A.2 PROOF TO PROP. 2

**Proposition 2.** *Let* $\mathbf{H}(\mathbf{w})$ *be the Hessian matrix for the model with parameter* $\mathbf{w}$*. With the definition of eigenvector* $\mathbf{s}$ *and greatest eigenvalue* $\rho$ *of* $\mathbf{H}(\mathbf{w})\mathbf{s} = \rho\mathbf{s}$*, there lies a positive correlation of* $\rho \propto g$*.*

*Proof.* To prove the above proposition, we need to expand the conclusion of Thm. 3 to all model layers. According to the chain gradient, the forward process can be denoted as $\mathbf{x}_{(i+1)} = \mathbf{w}_{(i)}^{\mathrm{T}}\mathbf{x}_{(i)} + b_{(i)}$. Thus, the gradient is represented as $\mathbf{w}_{(i)}^{\mathrm{T}} \cdot \mathbf{g}_{(i)}$. Given two logits gradients of $\mathbf{g}$ for ERM and $\widehat{\mathbf{g}}$ for

DuRM, we have $\widehat{\mathbf{g}} = \mathbf{g} + \mathbf{g}_d$ according to Thm. 3. Then, the gradient for penultimate layer fraction between DuRM and ERM can be derived as assuming it is bigger than 1 at first:

$$\frac{\mathbb{D}_w^2 \cdot (\mathbb{D}_g + \mathbb{D}_d)^2}{\mathbb{D}_w^2 + (\mathbb{D}_g + \mathbb{D}_d)^2} \cdot \frac{\mathbb{D}_w^2 + \mathbb{D}_g^2}{\mathbb{D}_w^2 \cdot \mathbb{D}_g^2} \geq 1$$

$$\frac{(1 + \frac{\mathbb{D}_d}{\mathbb{D}_g})^2}{1 + \frac{2\mathbb{D}_g \mathbb{D}_d + \mathbb{D}_d^2}{\mathbb{D}_w^2 + \mathbb{D}_g^2}} \geq 1$$

$$\frac{2\mathbb{D}_d}{\mathbb{D}_g} + \frac{\mathbb{D}_d^2}{\mathbb{D}_g^2} \geq \frac{2\mathbb{D}_d \mathbb{D}_g + \mathbb{D}_d^2}{\mathbb{D}_w^2 + \mathbb{D}_g^2}$$

$$\mathbb{D}_w^2 + \mathbb{D}_g^2 \geq \mathbb{D}_g^2, \tag{19}$$

which obviously holds. Thus, the assumption also holds. The Eq. equation 19 shows the consistency properties between the logits gradient with model gradient.

By now, we are ready to prove the above proposition. Given the Hessian matrix of $\mathbf{H}(w)$, the eigenvector $s$ can be defined as $\mathbf{H}(\mathbf{w})\mathbf{s} = \rho\mathbf{s}$, where $\rho$ is its corresponding greatest eigenvalue. Recall that the gradient descent (GD) can be formulated as:

$$\mathbf{w} + \mathbf{v} = \mathbf{w} - \nabla\ell(w), \tag{20}$$

where $\mathbf{v}$ denotes one weight step after one GD step. The updated weight is thus represented as $\mathbf{w} + \mathbf{v}$. Then, let us apply a two-order Taylor expansion to the loss function of $\mathbf{w} + \mathbf{v}$:

$$\ell(\mathbf{w} + \mathbf{v}) = \ell(\mathbf{w}) + \mathbf{v}^{\mathrm{T}}\mathbf{J}(\mathbf{w}) + \frac{1}{2}\mathbf{v}^{\mathrm{T}}\mathbf{H}(\mathbf{w})\mathbf{v} + o(\|\mathbf{v}\|^2). \tag{21}$$

Based on Eq. equation 21, a smaller $\ell(\mathbf{w} + \mathbf{v})$ could be achieved when the right hand term is less than or equal to zero. Then, when we pick $\mathbf{s}$ as DG direction, let us represent weight step into the eigenvector with greatest eigenvalue of Hessian matrix as $\|\mathbf{v}\| \cdot \mathbf{s}$. Such a process can be formulated as:

$$\mathbf{v}^{\mathrm{T}}\mathbf{J}(\mathbf{w}) + \frac{1}{2}\mathbf{v}^{\mathrm{T}}\mathbf{H}(\mathbf{w})\mathbf{v} \leq 0$$

$$\mathbf{s}^{\mathrm{T}} \cdot \mathbf{J}(\mathbf{w}) + \frac{1}{2}\rho \cdot \mathbf{s}^{\mathrm{T}} \cdot \mathbf{s} \leq 0$$

$$\mathbf{s}^{\mathrm{T}} \cdot \mathbf{J}(\mathbf{w}) + \frac{1}{2}\rho \cdot 1 \leq 0$$

$$\left\|\mathbf{s}^{\mathrm{T}} \cdot \mathbf{J}(\mathbf{w})\right\| \cdot \frac{\mathbf{s}^{\mathrm{T}}}{\|\mathbf{s}^{\mathrm{T}}\|} \cdot \frac{\mathbf{J}(\mathbf{w})}{\|\mathbf{J}(\mathbf{w})\|} + \frac{1}{2}\rho \leq 0 \tag{22}$$

Later, when $\rho$ is larger, the gradient projection on $\mathbf{s}$ becomes larger, which makes gradient larger. $\square$

### A.3 PROOF TO THM. 4

**Theorem 4.** *Given gradients $g \sim \mathcal{N}(\mu_1, \sigma_1^2)$ and $\widehat{g} \sim \mathcal{N}(\mu_2, \sigma_2^2)$, where $\mu_1 = \mu_2$ and $\sigma_1 < \sigma_2$. Assume gradient descent has $T$ steps. The inequality between minimum order statistics of empirical gradients holds with a high probability of $P(g_{(1)} \geq \widehat{g}_{(1)}) = \frac{1}{2} + T \int [1 - F_g(\widehat{g})]^T - [1 - F_{\widehat{g}}(\widehat{g})]^T \cdot (1 - F_{\widehat{g}_{(1)}})^{T-1} \mathrm{d}F_{\widehat{g}_{(1)}}$, such a probability owns a numerical solution of $0.5 + \Theta(\frac{\sigma_1}{\sigma_2})$, where $\Theta(\frac{\sigma_1}{\sigma_2}) \geq 0, \forall \sigma_1 < \sigma_2 \in \mathbb{R}$.*

*Proof.* Given two minimal order statistics gradient of $g_{(1)}$ for ERM and $\widehat{g}_{(1)}$, the two models with the same initialization state are updated via $N$ steps to sample $N$ gradients. Moreover, we have $g \sim \mathcal{N}(\mu_1, \sigma_1^2)$ and $\widehat{g} \sim \mathcal{N}(\mu_2, \sigma_2^2)$, where $\mu_1 = \mu_2$ and $\sigma_1 < \sigma_2$. The probability density function and cumulative distribution function of minimal order statistics to a distribution variable can be

derived as:

$$
\begin{aligned}
F_{g_{(1)}}(g) = P(g_{(1)} \leq g) &= 1 - P(g_{(1)} > g) \\
&= 1 - P(g_1 > g, g_2 > g, \cdots, g_N > g) \\
&= 1 - P(g_1 > g) \cdot P(g_2 > g) \cdots P(g_N > g) \\
&= 1 - [1 - F(g)]^N
\end{aligned}
$$

$$
f_{g_{(1)}}(g) = \frac{\mathrm{d}F_{g_{(1)}}(g)}{\mathrm{d}g} = N \cdot [1 - F(g)]^{N-1} \cdot f(g). \tag{23}
$$

Then, we can derive the probability of $P(g_{(1)} > \widehat{g}_{(1)})$ of Eq. equation 24:

$$
\begin{aligned}
P(g_{(1)} > \widehat{g}_{(1)}) = P[I(g > \widehat{g})|\widehat{g}] &= \int P(g_{(1)} > \widehat{g}_{(1)}|\widehat{g}_{(1)} = \widehat{g}) \cdot f_{\widehat{g}_{(1)}}(\widehat{g})\mathrm{d}\widehat{g} \\
&= \int [1 - \int_{-\infty}^{g} f_{g_{(1)}}(g)\mathrm{d}g] \cdot f_{\widehat{g}_{(1)}}(\widehat{g})\mathrm{d}\widehat{g} \\
&= \int N[1 - F_g(\widehat{g})]^N \cdot [1 - F_{\widehat{g}}(\widehat{g})]^{N-1} \cdot f_{\widehat{g}}(\widehat{g})\mathrm{d}\widehat{g} \\
&= N \int_0^1 [1 - F_g(\widehat{g})]^N \cdot [1 - F_{\widehat{g}}(\widehat{g})]^{N-1} \cdot \mathrm{d}F_{\widehat{g}}(\widehat{g}) \\
&= N \int_0^1 [1 - F_{\widehat{g}}(\widehat{g})]^{2N-1}\mathrm{d}F_{\widehat{g}}(\widehat{g}) + N \int_0^1 \Delta F \mathrm{d}F_{\widehat{g}}(\widehat{g}) \\
&= \frac{1}{2} + N \int_0^1 \Delta F \mathrm{d}F_{\widehat{g}}(\widehat{g}) \geq \frac{1}{2};
\end{aligned}
\tag{24}
$$

in which $\Delta F$ in Eq. equation 24 has been represented by and has an inequality of Eq. equation 25:

$$
\Delta F = [1 - F_g(\widehat{g})]^N - [1 - F_{\widehat{g}}(\widehat{g})]^N. \tag{25}
$$

At next, we need to calculate the term of $N \int_0^1 \Delta F \mathrm{d}F_{\widehat{g}}(\widehat{g})$. Due to the existence of the CDF of Gaussian distribution, this term could not be derived an analytical solution. To this end, we obtain its numerical solution as a number always larger than 0. To prove that, we conduct analysis from two perspective. On one hand, as shown in Figure 6, when a reversed distribution with larger variance subtracts another distribution, such a value is lower than 0 on the left side of $\mu$, which is greater than 0 on the right side of $\mu$. What's more, when multiplying this scalar with the differentiation of CDF, which is growing larger as $x$-axis, the final summed results is always greater than 0. On the other hand, we simulate the numerical solution of Figure 9(c). We firstly divide $N \int_0^1 [1 - F_{\widehat{g}}(\widehat{g})]^{2N-1}\mathrm{d}F_{\widehat{g}}(\widehat{g}) + N \int_0^1 \Delta F \mathrm{d}F_{\widehat{g}}(\widehat{g})$ into two terms, which are the fixed one $N \int_0^1 [1 - F_{\widehat{g}}(\widehat{g})]^{2N-1}\mathrm{d}F_{\widehat{g}}(\widehat{g})$ and the varying one of $N \int_0^1 \Delta F \mathrm{d}F_{\widehat{g}}(\widehat{g})$. As the result shown, we have the above results and theorem holds. $\square$

## B  IMPLEMENTATION DETAILS

### B.1  CONVENTIONAL CLASSIFICATION

For conventional classification tasks, we implement experiments of ImageNet-1KDeng et al. (2009) on one NVIDIA A100 with a total memory of 80G. For dataset preparation details, an image resolution of 224, an augmentation strategy of RandAug are utilized. For training details, a batch size of 512, a learning rate of 0.05, a rate decay of Step Scheduler, an optimizer of SGD along with momentum and $\ell_2$ regularization, an epoch of 90 are utilized. For downstream vision classification datasets, we implement experiments on several NVIDIA Tesla V100 with memory below 16G. For dataset preparation, the CIFAR-10Krizhevsky et al. (2009) and CIFAR-100Krizhevsky et al. (2009) have resolution of 32 and others vision have resolution of 224. For training details, a batch size of 128, a learning rate of 0.01, an optimizer of SGD along with momentum and $\ell_2$ regularization, an epoch of 120 are adopted. For classification on THUCNews and UrbanSound8K, we follow the implementation of Li & Sun (2007) and Salamon & Bello (2017).

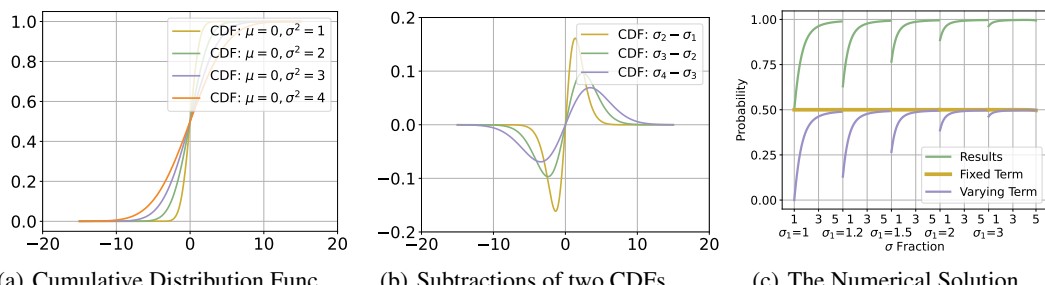

(a) Cumulative Distribution Func.  (b) Subtractions of two CDFs.  (c) The Numerical Solution

Figure 6: (a) The CDF of four kinds of Gaussian distributions with different variance. (b) The distribution of subtractions between two reversed CDF-s $(1 - F(g))$, in which the distribution with bigger variance is as the minuend. (c) The numerical solution of Eq. equation 24. The Eq. equation 24 can be divided into two terms, namely the fixed one and the varying one. Then, the final results are recorded in the figure.

**Adversarial Robustness**    For training details, we utilize the same parameters to the experiments to conventional classification. Concretely, for FGSM setting, we follow the work in Goodfellow et al. (2015). For PGD setting, we follow the work in Carlini & Wagner (2017).

**Long-tailed Recognition**    For training details, we utilize the same parameters to the experiments to conventional classification. As for dataset settings, we totally follow the works in Tang et al. (2020).

### B.2 SEMANTIC SEGMENTATION

The semantic segmentation tasks are implemented on one NVIDIA A100 with a total memory below 80G. And all of our implementations are following Contributors (2020) default with only changing the number of output dimension to deploy `DuRM`. Concretely, for FCN-ResNet-101Long et al. (2015), HRNet-48Wang et al. (2020) and PSPNet-ResNet18Zhao et al. (2017), we utilize a crop size of $512 \times 1024$. An optimizer of SGD with a learning rate of 0.01, a momentum of 0.9 and a weight decay of 0.0005. The *PolyLR* learning rate decay is also adopted with a minimum of 0.0001, a power of 0.9. Then, we train the model with 40000 iteration, among which the validation is deployed for every 4000 iterations. For Deeplab-ResNet101Chen et al. (2017), a crop size of $512 \times 512$ is leveraged. For Segformer B0 and B5Xie et al. (2021), a crop size of $1024 \times 1024$ is utilized. An optimizer of AdamW along with learning rate of 0.00006, betas of (0.9, 0.99) and a weight decay of 0.01 are adopted. Then, we use a *LinearLR* for learning rate decay with a start factor of 1.6. Later, the models are trained with 160000 iterations.

### B.3 OOD GENERALIZATION

The experiments are implemented on one NVIDIA V100 with a total memory below 16G. Following the task setting of Gulrajani & Lopez-Paz (2020), each dataset is composed of four different domains. We run four times for one dataset to realize one domain for testing while the left three for training. For dataset preparation details, an image resolution of 224, an augmentation of RandAug are utilized. For training details, a batch size of 64, a learning rate of 0.01, an optimizer of SGD along with momentum, an epoch of 90 for each setting are utilized. The model selection is based on validation set divided from training set.

## C   EXPERIMENTAL RESULTS

### C.1   DETAILED CLASSIFICATION RESULTS

In this subsection, we array the whole classification results of our three versions of `DuRM` in Table 8. Concretely, we compare `DuRM` with three kinds of dummy class settings varying from 1 to 3. Then, the other experimental settings are kept as the same as Table 1, in which only `DuRM-2` is conducted to verify the effectiveness the the devised `DuRM`. As shown in Table 8, our `DuRM` can outperform

Table 8: Classification accuracy on different datasets. ERM is the compared method, while DuRM-x is the different realize to our method, in which the x denotes counts of dummy class.

| Datasets | Model | ERM (%) | DuRM-1(%) | DuRM-2(%) | DuRM-3(%) | Improvement (%) |
|---|---|---|---|---|---|---|
| ImageNet-1K | ResNet-18 | 69.74±0.09 | **69.87**±0.06 | 69.76±0.06 | 69.68±0.11 | 0.13 / 0.02 /-0.06 |
| THUCNews | Transformer | 89.55±0.46 | **90.08**±0.27 | 89.92±0.31 | 90.02±0.13 | 0.53 / 0.37 / 0.47 |
| UrbanSound8K | LSTM | 65.16±0.08 | 66.13±0.17 | 66.00±1.01 | 64.96±1.18 | 0.97 / 0.84 /-0.20 |
| CIFAR-10 | ResNet-18 | 82.08±0.34 | **82.47**±0.28 | 82.11±0.30 | 82.31±0.27 | 0.39 / 0.03 / 0.23 |
|  | ResNet-50 | 84.24±0.22 | 84.22±0.14 | 84.10±0.57 | **84.25**±0.32 | -0.02 /-0.14 / 0.01 |
|  | ViT-Tiny | 92.79±0.67 | 92.55±0.86 | **94.38**±0.44 | 93.77±0.31 | -0.24 / 1.59 / 0.98 |
| CIFAR-100 | ResNet-18 | 58.42±0.33 | 58.15±0.09 | **58.50**±0.38 | 58.29±0.03 | -0.27 / 0.08 /-0.13 |
|  | ResNet-50 | 60.17±2.06 | 58.32±1.28 | **61.03**±0.11 | 60.33±1.56 | -1.85 / 0.86 / 0.16 |
|  | ResNext-50 | 65.02±0.35 | 64.90±0.24 | 64.44±0.10 | **65.32**±0.20 | -0.12 /-0.58 / 0.30 |
| STL-10 | ResNet-18 | 85.20±0.75 | 85.56±1.02 | 86.21±0.73 | **86.55**±0.38 | 0.45 / 1.01 / 1.35 |
|  | Swin-Tiny | 95.55±0.47 | 95.56±0.69 | **96.33**±0.14 | 95.61±0.25 | 0.01 / 0.78 / 0.06 |
|  | MobileViT-S | 94.32±0.15 | 95.20±0.49 | 95.03±0.53 | **95.31**±0.71 | 0.88 / 0.71 / 0.99 |
| Oxford-Pet | ResNet-18 | 90.23±0.19 | 90.13±0.30 | 90.30±0.21 | **90.38**±0.27 | -0.10 / 0.07 / 0.15 |
|  | ResNext-50 | 93.51±0.16 | 93.57±0.18 | 93.57±0.17 | **93.58**±0.04 | 0.06 / 0.06 / 0.07 |
|  | MLP-Mixer-B | 85.06±0.33 | **85.62**±0.29 | 84.71±0.47 | 85.52±0.62 | 0.56 /-0.35 / 0.46 |
| Flower-102 | ResNet-18 | 85.07±0.24 | 84.88±0.21 | 85.08±0.19 | **85.15**±0.13 | -0.19 / 0.01 / 0.08 |
|  | ResNet-101 | 86.69±0.23 | 86.63±0.03 | **86.91**±0.12 | 86.72±0.11 | -0.06 / 0.22 / 0.03 |
|  | ViT-Tiny | 87.33±1.56 | 86.95±1.47 | 88.71±1.17 | **89.13**±0.25 | -0.38 / 1.38 / 1.80 |

ERM on all datasets with different backbone models. Among them, the greatest achievement 1.38% is obtained by experiments on dataset of Flower-102Nilsback & Zisserman (2008) with a backbone of ViT-TinyDosovitskiy et al. (2021) and DuRM-2. Please notice that our `DuRM` is almost a free lunch method. Thus, to achieve such an improvement is non-trivial.

## C.2 DETAILED SEMANTIC SEGMENTATION RESULTS

In this subsection, we array the whole semantic segmentation results for each category, namely class-wise IoU along with the class-mean IoU (mIoU). Concretely, the category level results on CityScapesCordts et al. (2016) with 19 classes and Pascal VOC-2012Everingham et al. (2015) with 22 classes are listed as shown in Table 9 and Table 10. Concretely, as shown in Table 9, the results tell us that `DuRM` is inclined to be more robust to rare class or hard samples, which is reflected as that `DuRM` achieves an IoU of 47.01 on *Truck* via FCN-ResNet101 comparing with an IoU of 10.03 achieved by ERM under the same setting. Moreover, there seems not to have the same phenomenon in the failure cases of `DuRM`. Concretely, when `DuRM` fails to outperform ERM on CityScapes, such a degradation is smaller than improvement bu `DuRM`, i.e., `DuRM` degrades IoU on *Pole* with FCN-ResNet101, while this degradation is 1.28%, being greatly smaller than the aforementioned improvement by 36.98%. As for the results in Table 10, it shows that even our `DuRM` fails to outperform ERM on validation set, we still achieve a greater mIoU on test set, which is more vital and further demonsrtates `DuRM` aids model to achieve a better generalization state.

## C.2.1 LOVEDA RESULTS

In this subsection, we array the detailed results of LoveDAWang et al. (2021) dataset. Concretely, we implement Segmentor-LargeStrudel et al. (2021) and UperNet-SwinTransformerWang et al. (2023) methods on it which are two popular semantic segmentation paradigm.

What interesting is the Segmentor-Large with `DuRM` predict some pixels into the dummy class. To further explore the phenomenon, we make some visualization on LoveDA prediction with some dummy class predictions. To achieve accurate analysis, we seek for the author of LoveDA to obtain two ground truth masks. As shown in Figure 7, a red box annotates the dummy class region. As we can see in image *4700*, the dummy class marks a small portion of mis-labeled pixels by two models (ERM and `DuRM`), which means `DuRM` could facilitate exploiting novel knowledge. As for

Table 9: The detailed category level semantic segmentation results on CityScapes, in which the 19 classes are included.

| Method | Road | Sidew. | Buid | Wal | Fence | Pole | Ligt | sign | Vega. | Ter. | sky | Person | Rider | Car | Truck | Bus | Train | Motor | Bicycle | mIoU |
|---|---|---|---|---|---|---|---|---|---|---|---|---|---|---|---|---|---|---|---|---|
| | | | | | | | | | Validation Set | | | | | | | | | | | |
| FCN-R101 | 97.85 | 82.41 | 90.3 | 25.36 | 45.08 | 62.32 | 67.15 | 76.27 | 91.36 | 53.62 | 93.79 | 79.49 | 57.51 | 91.75 | 10.03 | 57.66 | 11.67 | 48.74 | 74.61 | 64.05 |
| +DuRM | 97.65 | 81.86 | 89.95 | 23.30 | 42.29 | 61.03 | 67.36 | 76.13 | 91.60 | 54.50 | 93.97 | 78.92 | 52.77 | 92.93 | 47.01 | 49.61 | 13.25 | 47.21 | 74.73 | 65.05 |
| Mit-B0 | 97.86 | 82.87 | 91.17 | 57.16 | 52.33 | 56.53 | 63.93 | 73.02 | 92.10 | 63.07 | 94.37 | 77.65 | 52.04 | 93.49 | 75.99 | 77.78 | 66.38 | 56.58 | 73.4 | 73.57 |
| +DuRM | 98.05 | 84.03 | 91.91 | 59.14 | 55.02 | 59.63 | 66.90 | 75.47 | 92.39 | 65.3 | 94.80 | 79.10 | 54.90 | 94.24 | 77.22 | 83.21 | 75.61 | 61.93 | 75.00 | 75.99 |
| Mit-B5 | 98.55 | 87.96 | 93.66 | 67.93 | 65.62 | 68.91 | 74.50 | 81.13 | 93.26 | 67.19 | 95.68 | 84.55 | 67.07 | 95.50 | 81.90 | 92.20 | 86.56 | 72.83 | 79.47 | 81.82 |
| +DuRM | 98.55 | 87.86 | 93.75 | 68.43 | 66.98 | 68.78 | 74.55 | 81.45 | 93.12 | 64.77 | 95.64 | 84.51 | 67.80 | 95.83 | 88.58 | 91.32 | 84.42 | 73.59 | 79.65 | 82.08 |
| | | | | | | | | | Test Set | | | | | | | | | | | |
| FCN-R101 | 97.99 | 81.88 | 89.87 | 28.43 | 42.20 | 60.12 | 69.81 | 72.37 | 92.19 | 65.96 | 94.24 | 81.13 | 56.04 | 91.48 | 3.82 | 31.42 | 5.4 | 32.35 | 68.98 | 61.35 |
| +DuRM | 97.85 | 80.62 | 89.61 | 19.71 | 41.63 | 59.49 | 68.33 | 72.04 | 92.17 | 64.55 | 93.98 | 81.06 | 52.92 | 92.99 | 32.68 | 30.28 | 9.14 | 55.53 | 69.06 | 63.35 |
| Mit-B0 | 98.00 | 82.34 | 91.27 | 51.31 | 51.23 | 53.86 | 64.02 | 70.64 | 92.57 | 69.62 | 95.08 | 80.83 | 59.12 | 94.14 | 66.11 | 78.62 | 72.63 | 56.83 | 69.36 | 73.56 |
| +DuRM | 98.19 | 83.36 | 91.84 | 49.57 | 54.77 | 57.59 | 67.04 | 72.58 | 92.69 | 69.83 | 95.30 | 81.92 | 61.16 | 94.67 | 64.23 | 74.75 | 69.52 | 60.10 | 71.17 | 74.23 |
| Mit-B5 | 98.59 | 86.33 | 93.61 | 56.68 | 62.82 | 68.14 | 74.83 | 79.29 | 93.54 | 72.60 | 95.84 | 86.56 | 70.58 | 95.85 | 73.14 | 88.86 | 87.56 | 71.18 | 76.43 | 80.66 |
| +DuRM | 98.66 | 86.75 | 93.64 | 55.30 | 63.06 | 68.09 | 75.26 | 79.09 | 93.57 | 72.42 | 95.84 | 86.58 | 70.94 | 95.91 | 76.25 | 90.39 | 87.74 | 70.99 | 76.32 | 80.88 |

Table 10: The detailed category level semantic segmentation results on Pascal VOC2012, in which the 22 classes are included.

| Method | back. | aero. | bicycle | bird | boat | bottle | bus | car | cat | chair | cow | table | dog | horse | motor. | person | plant | sheep | sofa | train | monitor | mIoU |
|---|---|---|---|---|---|---|---|---|---|---|---|---|---|---|---|---|---|---|---|---|---|---|
| | | | | | | | | | | Validation Set | | | | | | | | | | | | |
| DLB-R101 | 92.32 | 85.43 | 44.57 | 67.88 | 55.49 | 50.32 | 81.00 | 78.52 | 79.82 | 27.51 | 67.35 | 32.88 | 62.65 | 72.05 | 76.96 | 80.34 | 44.85 | 72.23 | 37.67 | 72.46 | 55.83 | 63.72 |
| +DuRM | 92.58 | 78.46 | 51.55 | 65.99 | 53.01 | 53.32 | 74.42 | 77.57 | 77.57 | 25.90 | 63.34 | 52.66 | 64.63 | 74.17 | 80.01 | 81.34 | 53.65 | 46.33 | 42.14 | 71.86 | 53.89 | 63.54 |
| HRNet-W48 | 91.03 | 55.52 | 58.24 | 68.91 | 57.00 | 52.03 | 51.43 | 64.36 | 70.53 | 22.15 | 59.43 | 35.41 | 61.68 | 68.08 | 55.35 | 79.28 | 34.29 | 68.05 | 36.25 | 62.78 | 56.32 | 57.53 |
| +DuRM | 91.47 | 77.29 | 57.50 | 66.42 | 34.68 | 53.82 | 78.64 | 77.15 | 52.52 | 25.21 | 59.40 | 36.93 | 52.08 | 54.03 | 72.38 | 79.75 | 40.42 | 64.10 | 35.87 | 68.77 | 64.00 | 59.16 |
| PSPNet-R18 | 90.32 | 75.47 | 49.14 | 52.46 | 51.84 | 47.57 | 80.22 | 73.04 | 68.54 | 15.55 | 56.71 | 48.16 | 53.09 | 61.91 | 72.73 | 72.08 | 33.21 | 64.57 | 27.59 | 70.50 | 50.54 | 57.87 |
| +DuRM | 90.10 | 75.81 | 44.79 | 56.42 | 54.53 | 46.91 | 81.44 | 70.40 | 69.07 | 17.59 | 53.94 | 47.29 | 55.74 | 57.17 | 73.01 | 73.09 | 29.85 | 68.94 | 26.97 | 73.14 | 51.46 | 57.98 |
| | | | | | | | | | | Test Set | | | | | | | | | | | | |
| DLB-R101 | 91.98 | 79.02 | 45.48 | 62.74 | 50.93 | 53.72 | 83.89 | 71.51 | 71.03 | 21.16 | 74.26 | 29.73 | 60.58 | 80.47 | 78.84 | 79.23 | 46.5 | 65.13 | 36.35 | 70.72 | 39.25 | 61.55 |
| +DuRM | 92.39 | 76.04 | 51.46 | 72.84 | 44.63 | 52.57 | 77.73 | 73.08 | 75.73 | 23.63 | 62.79 | 53.32 | 64.66 | 66.64 | 79.60 | 78.31 | 48.44 | 52.03 | 41.35 | 58.41 | 48.74 | 61.64 |
| HRNet-W48 | 91.48 | 52.37 | 54.80 | 68.96 | 44.15 | 60.95 | 53.14 | 69.88 | 72.93 | 20.59 | 54.41 | 41.00 | 64.79 | 65.89 | 51.61 | 78.35 | 40.25 | 65.63 | 42.61 | 68.23 | 51.35 | 57.78 |
| +DuRM | 92.00 | 74.01 | 52.17 | 62.68 | 31.02 | 63.69 | 79.00 | 76.83 | 51.60 | 23.46 | 62.43 | 41.09 | 53.97 | 57.95 | 70.70 | 77.48 | 47.15 | 69.05 | 43.40 | 68.13 | 54.35 | 59.63 |
| PSP-R18 | 89.92 | 70.47 | 47.80 | 59.10 | 39.85 | 52.33 | 78.05 | 70.58 | 69.95 | 11.36 | 55.57 | 51.16 | 57.19 | 56.29 | 73.97 | 69.61 | 31.57 | 58.57 | 32.03 | 64.70 | 42.13 | 56.29 |
| +DuRM | 90.06 | 70.82 | 46.28 | 65.5 | 40.37 | 54.28 | 74.37 | 73.20 | 65.50 | 17.10 | 54.26 | 51.51 | 53.92 | 61.22 | 75.03 | 68.94 | 31.09 | 60.35 | 32.27 | 60.83 | 49.23 | 56.96 |

image *4409*, the segmentor obviously fails to segment the *water* region , where some uncertain pixels are segmented into the dummy class. To this end, the dummy class could also be an indicator on measuring the model confidence or prediction credibility.

## C.3 DETAILED OOD GENERALIZATION RESULTS

In this subsection, we list the detailed performance of our `DuRM` plugged in the various state-of-the-art OOD generalization methods. Concretely, we widely validate these methods on four mainstream OOD generalization benchmark, namely VLCSFang et al. (2013), PACSLi et al. (2017), OfficeHomeVenkateswara et al. (2017) and TerraIncBeery et al. (2018). Among these benchmarks, four domains are included within each benchmark. To better validate `DuRM` performance, we list the testing accuracy on each domain for all adopted benchmarks in Table 11. As the results shown, there goes with clear evidence that `DuRM` facilitates obtaining a better generalization.

## C.4 DETAILED ADVERSARIAL ROBUSTNESS RESULTS

In this subsection, we validate how `DuRM` performs under the two kinds of mainstream adversarial attacking scenarios. Concretely, as shown in Table 12, we test FGSMGoodfellow et al. (2015) and PGDCarlini & Wagner (2017) attack to a ResNet-18 model on CIFAR-10 dataset. Then, we also conduct adversarial training to cope with the corresponding attack. As the results shown, our proposed `DuRM` comprehensively outperform ERM on various task settings.

## C.5 DETAILED LONG-TAILED RECOGNITION RESULTS

In this subsection, we further conduct `DuRM` on three kinds of long-tailed classification scenarios. Concretely, following the previous works, we manually construct long-tailed CIFAR-10 with imbalanced ratios of 100, 50, and 10, among which 100 is the hardest settings and 10 is the easiest setting. As shown in Table 13, `DuRM` shows better generalization on long-tailed recognition scenarios. Moreover, we observe that `DuRM` performs better under more hard scenes.

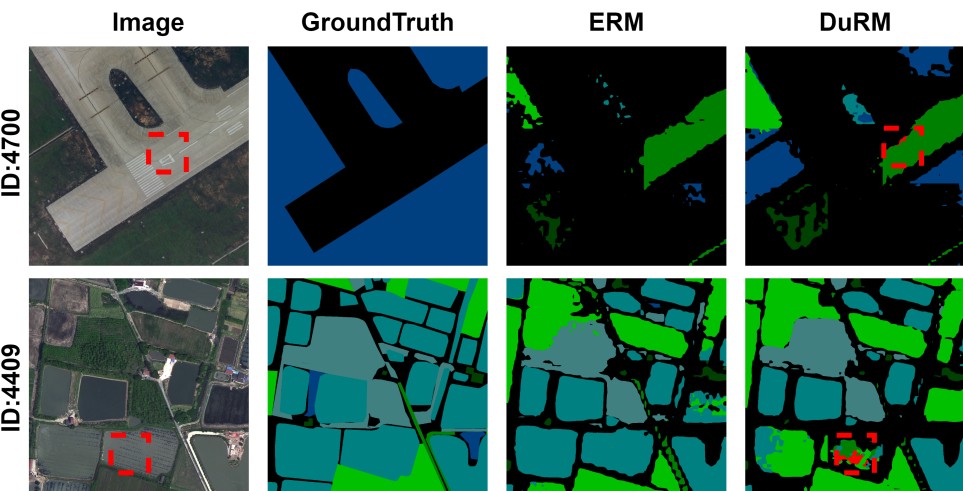

Figure 7: The visualization of results on LoveDA with dummy class output. The models are trained with a segmentor-L. The dummy class is marked in red, while other normal classes are marked in the same color among figures. Best view in color.

Table 11: The extensive domain level accuracy. By coincidence, all of these four benchmarks are composed of four different domains. Thus, the results when each domain is as testing set with three others as training are all listed. Finally, the results among four domains are averaged.

| Benchmark | Method | C | L | S | V | Average | Benchmark | Method | A | C | P | R | Average |
|---|---|---|---|---|---|---|---|---|---|---|---|---|---|
| VLCS | Vanilla | **94.40** | **62.44** | 65.26 | **67.58** | **72.42** | OfficeHome | Vanilla | **55.51** | 50.68 | 71.13 | 73.27 | 62.65 |
| | DuRM | 92.16 | 61.60 | **66.65** | 67.33 | 71.93 | | DuRM | 55.27 | **51.20** | **71.73** | **73.50** | **62.92** |
| | SWAD | 97.11 | 61.19 | 70.93 | 74.08 | 75.83 | | SWAD | 54.60 | 49.10 | 70.96 | 72.75 | 61.85 |
| | DuRM | **97.41** | **61.76** | **71.00** | **75.82** | **76.50** | | DuRM | **54.63** | **50.00** | **71.00** | 72.65 | **62.07** |
| | DANN | 92.30 | **62.57** | 61.52 | **61.25** | 69.41 | | DANN | 52.84 | 46.74 | 68.88 | **69.05** | 59.38 |
| | DuRM | **92.65** | 62.33 | **62.26** | 61.11 | **69.59** | | DuRM | **52.96** | **47.70** | **69.13** | 68.99 | **59.70** |
| | VRex | 95.45 | **62.75** | 69.55 | 72.86 | **75.15** | | VRex | **56.82** | 48.85 | **72.55** | 74.89 | 63.28 |
| | DuRM | **95.62** | 61.76 | **69.87** | 72.43 | 74.92 | | DuRM | 56.57 | **49.66** | 72.54 | **75.07** | **63.46** |
| | RSC | 95.22 | **64.02** | 68.77 | 71.49 | 74.87 | | RSC | 55.19 | 47.30 | 70.37 | 72.32 | 61.30 |
| | DuRM | 94.30 | 63.23 | **70.81** | 72.44 | **75.19** | | DuRM | **55.23** | **48.09** | **70.62** | **72.44** | **61.60** |
| | MMD | 95.85 | 60.72 | **70.97** | 70.03 | 74.39 | | MMD | **57.53** | **49.92** | **72.30** | **74.10** | **63.46** |
| | DuRM | **97.24** | **60.82** | 70.81 | **71.71** | **75.15** | | DuRM | 56.37 | 49.91 | 72.19 | 73.99 | 63.12 |

| Benchmark | Method | A | C | P | S | Average | Benchmark | Method | L100 | L38 | L43 | L46 | Average |
|---|---|---|---|---|---|---|---|---|---|---|---|---|---|
| PACS | Vanilla | 79.87 | 76.80 | **92.77** | 77.67 | 81.78 | TerraInc | Vanilla | 36.97 | 49.69 | **35.78** | 44.20 | **41.66** |
| | DuRM | **80.36** | **77.49** | 92.59 | 76.98 | **81.85** | | DuRM | 34.43 | **49.89** | 34.57 | 43.93 | 40.71 |
| | SWAD | **83.67** | 76.17 | 95.41 | 76.71 | 82.99 | | SWAD | 46.35 | 33.31 | 53.79 | 33.92 | 41.84 |
| | DuRM | 83.24 | **76.39** | **95.71** | **77.54** | **83.22** | | DuRM | 49.27 | **33.59** | **53.87** | **34.79** | **42.88** |
| | DANN | 70.36 | 73.86 | 88.68 | **79.03** | 77.98 | | DANN | 34.15 | **37.24** | 26.58 | **33.26** | **32.81** |
| | DuRM | **71.58** | **74.82** | **89.38** | 78.54 | **78.58** | | DuRM | **39.89** | 32.80 | 25.47 | 29.70 | 31.97 |
| | VRex | **80.24** | **76.19** | **95.51** | 72.60 | **81.13** | | VRex | 37.60 | **52.56** | 36.16 | **41.68** | 42.00 |
| | DuRM | 79.67 | 75.44 | 95.33 | **72.77** | 80.80 | | DuRM | **38.06** | 52.44 | **37.20** | 41.03 | **42.18** |
| | RSC | 80.31 | **75.98** | **93.37** | **74.19** | **80.96** | | RSC | **40.29** | **55.92** | 38.59 | 42.96 | 44.44 |
| | DuRM | 80.31 | 74.47 | 93.29 | 73.08 | 80.29 | | DuRM | 38.84 | 55.91 | **39.31** | **45.00** | **44.77** |
| | MMD | 80.63 | 74.12 | **92.26** | 77.40 | 81.10 | | MMD | 30.66 | 42.75 | **35.92** | 40.55 | 37.47 |
| | DuRM | **81.04** | **74.71** | 92.16 | **77.96** | **81.47** | | DuRM | **33.86** | **42.92** | 34.48 | 39.40 | **37.67** |

Table 12: Top-1 accuracy on adversarial robustness. The experiments are conducted on CIFAR-10 with a ResNet-18 model.

| Method | ERM | DuRM-1 | DuRM-2 | DuRM-3 |
|---|---|---|---|---|
| FGM Attack | 28.49 | 29.39 | **30.13** | 30.08 |
| FGM+Adv Training | 53.54 | **54.18** | 54.09 | 53.97 |
| PGD Attack | 12.55 | 13.29 | 13.09 | **13.79** |
| PGD + Adv Training | 46.85 | 46.89 | 46.65 | **47.10** |

Table 13: Classification accuracy on long-tailed CIFAR-10 datasets with a backbone model of ResNet-18.

| Imb. Ratio | Method (Top-1 Accuracy %) | | | |
|---|---|---|---|---|
| | ERM | DuRM-1 | DuRM-2 | DuRM-3 |
| 100 | 63.29 | 63.60 | 63.33 | **64.29** |
| 50 | 69.63 | **70.82** | 69.97 | 70.17 |
| 10 | 81.01 | 80.73 | **81.15** | 80.25 |

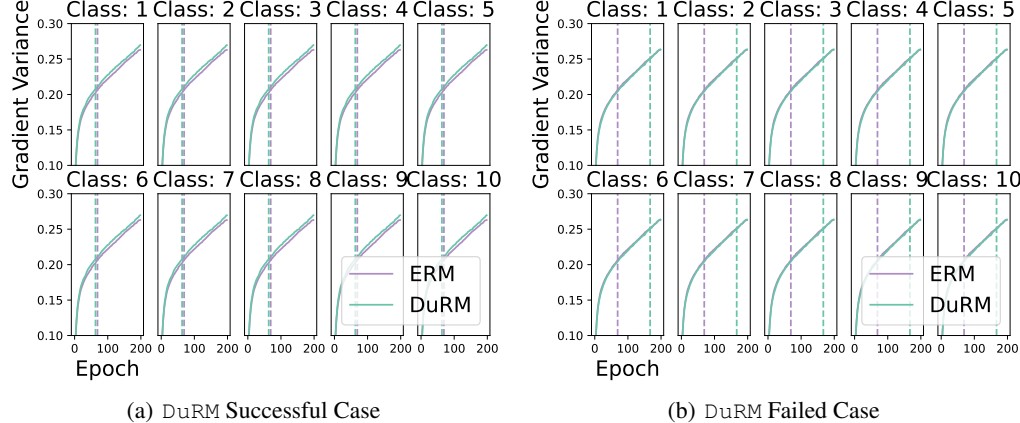

(a) DuRM Successful Case  (b) DuRM Failed Case

Figure 8: The convergence curve of training a ResNet-18 on CIFAR-10 with ERM, DuRM-1 (successful case) and DuRM-2 (failed case). We record the class-wise gradient variance and corresponding accuracy.

## C.6 SUPPLEMENTARY FOR GRADIENT VARIANCE

In this subsection, we conduct another failed case of DuRM on CIFAR-10 to compare the results of gradient variance. As shown in Figure 8, the successful case of DuRM achieves a greater gradient variance, while the failed case didn' t outperform ERM on gradient variance. What' s more, the accuracy values are not listed. Then we report here as: the failed case has an accuracy of 85.57%, the successful case has an accuracy of 84.76% and the ERM has an accuracy of 84.63%. Thus, the failed case can be blamed as the failure on improving gradient variance. To further support this claim, the dashed line mark where the best validated epoch appears. Furthermore, we notice that the successful case has better accuracy on all category, while the failed case has poor accuracy on all category.

## C.7 SUPPLEMENTARY FOR NUMBER OF DUMMY CLASS

In this subsection, we add the supplementary for the experiments in ablation study on the number of dummy class, as shown in Figure 9. Concretely, we additionally add experiments on another widely used dataset Oxford-Pet, which has more original number of class comparing with CIFAR-10 and STL-10. Despite that the count of DuRM outperforming ERM on Oxford-Pet becomes lower,

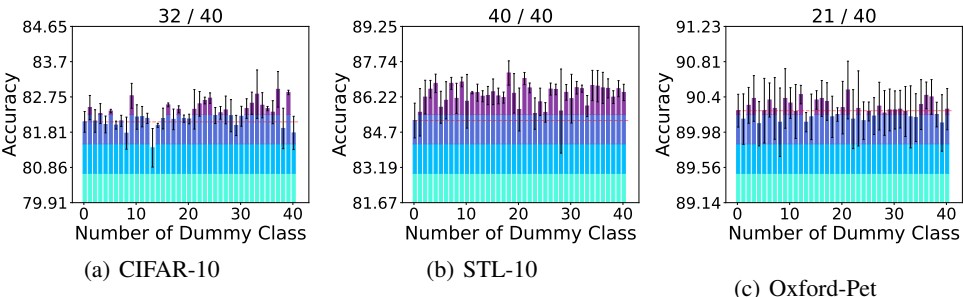

(a) CIFAR-10      (b) STL-10      (c) Oxford-Pet

Figure 9: The accuracy variance curve along the number of dummy class on CIFAR-10, STL-10 and Oxford-Pet.

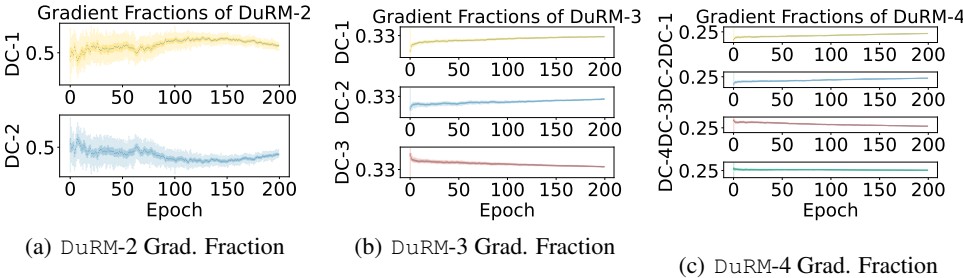

(a) DuRM-2 Grad. Fraction      (b) DuRM-3 Grad. Fraction      (c) DuRM-4 Grad. Fraction

Figure 10: The gradient fraction convergence curves for dummy classes under DuRM-2, DuRM-3 and DuRM-4.

there still goes without clear evidence that the number of dummy class is influencing the model generalization performance.

## C.8 Supplementary for Gradient Fraction

In this subsection, we add the supplementary for the experiments in analyzing the gradient fraction under multi-dummy class settings, including DuRM-2, DuRM-3 and DuRM-4, as shown in Figure 10. Concretely, we additionally add experiments on DuRM-4, in which the results further support our analysis in the main text that all the dummy classes are contributing to the corresponding gradient.

## C.9 Detailed Regularization Results

In this subsection, we provide the whole results about the comparison with other regularization methods, including Early Stop, L2, Momentum Gradient, EMA, MixUp and SWA. As shown in Table 14, our devised DuRM is able to co-operate with other regularization methods well and achieve better performance than simply applying them as a regularization. What' s more, our DuRM is easy enough, which makes us almost a free launch method.

Table 14: Compatibility to other regularization on CIFAR-10.

| Method | ERM | DuRM-1 | DuRM-2 | DuRM-3 |
|---|---|---|---|---|
| Vanilla | 79.82 | 79.86 | 79.75 | **80.01** |
| EarlyStop | 78.68 | 78.85 | 78.16 | **78.96** |
| L2 | 79.70 | 79.91 | **80.13** | 79.78 |
| Momentum | 81.98 | 82.05 | 82.22 | **82.33** |
| EMA | 79.62 | 79.65 | **79.99** | 79.92 |
| MixUp | 82.91 | 82.85 | **83.50** | 82.76 |
| SWA | 93.46 | **93.48** | 93.43 | **93.48** |

