# OpenReview forum: "Frustratingly Easy Model Generalization by Dummy Risk Minimization"
_ICLR.cc/2024/Conference — ICLR 2024 Conference Withdrawn Submission_

### Official Review · Reviewer_ARGr · 2023-10-24

**Soundness:** 3 good
**Presentation:** 3 good
**Contribution:** 2 fair
**Rating:** 5
**Confidence:** 4

**Summary:**

This paper proposes an interesting Dummy Risk Minimization (DuRM) paradigm for the fundamental ML training framework, which enlarges the dimension of the output logits and then optimizes with SGD. Some theoretical and empirical studies are presented to show the effectiveness of the proposed DuRM.

**Strengths:**

The proposed paradigm is interesting and might bring new insight to the ML community.

**Weaknesses:**

A significant weakness of this paper is that its theoretical analyses are limited. Specifically, the theoretical results are highly dependent on the strong assumption (i.e., Gaussian distribution), which is insufficient for a model. It is better to see a result related to the underlying model parameters or training process.

Another concern is the consistent attack manner for the adversarial robustness. In tab.5, are the models trained by ERM/DuRM attacked by their corresponding training loss? If true, what will happen if we use ERM+PGD/FGSM for DuRM? Besides, parameter-free AutoAttack should be further considered.

Moreover, one of the effective paradigms is AUC optimization [1,2,3], which is distribution-insensitive concerning long-tailed recognition. In light of this, how is the performance of DuRM compared with AUC? Then, can we apply a similar dummy trick to such a pairwise paradigm?

Ref:

[1] AUC Maximization in the Era of Big Data and AI: A Survey.

[2] When AUC meets DRO: Optimizing Partial AUC for Deep Learning with Non-Convex Convergence Guarantee.

[3] Learning with Multiclass AUC: Theory and Algorithms.

**Questions:**

Please see the Weaknesses above.

---

### Official Review · Reviewer_L9Sz · 2023-10-26

**Soundness:** 1 poor
**Presentation:** 2 fair
**Contribution:** 2 fair
**Rating:** 3
**Confidence:** 3

**Summary:**

This submission proposes to improve the generalization of ERM by adding "dummy classes" to a classification task. It claims that these dummy classes can provide gradient regularization, which helps find flatter minima and thus improves generalization. The experiments show that the proposed method can improve the performance on standard classification tasks, as well as adversarial robustness, long-tailed classification, and OOD generalization.

**Strengths:**

The proposed DuRM is indeed very simple. If it really works then could be interesting.

**Weaknesses:**

My biggest concern is that many claims in the submission are not very sound, and the proposed method is not sufficiently justified either theoretically or empirically. After reading the manuscript, I am not convinced at all that DuRM could really improve over ERM, except that it provides more hyperparameters so increases the degree of freedom for hyperparameter tuning. Thus, I recommend rejecting this submission.

Here are my detailed comments:
### 1. Regarding the usage of the term "generalization"
In ML there are two types of generalization, and I am not quite sure which one of them this work is talking about. The first type is train-to-test generalization, that is whether a model trained on the train set can perform well on the test set from the same distribution. The second type is OOD generalization, that is whether a model trained on one domain or distribution can perform well on another one. The introduction of this submission seems to focus on OOD generalization, yet the experiments also include standard classification tasks and segmentation tasks. Thus, I am not sure which generalization the authors want to claim that their method could improve. But either way, after reading the submission, I really cannot see how DuRM could improve either type of generalization.

### 2. How the proposed DuRM method is implemented is not very clear
First, I think Eqn. (2) is incorrect: The second $h_{erm}$ should be $h_{durm}$. Second, I couldn't find in the submission how this $h_{durm}$ is implemented, and there is no code attached. I assume that the authors add extra linear heads to the last linear layer for the dummy classes. Please let me know whether this is correct.

### 3. Regarding Section 2.2 about gradient regularization
The whole Section 2.2 is very confusing, and does not seem very sound:
- "No samples are classified into dummy classes": I guess the intuition here is that since no training sample belongs to the dummy classes, their weights will either converge to zero, or if used with softmax converge to a very negative point. But is it really true that absolutely not samples could be classified into the dummy classes? Not even under adversarial attack? For example, what would happen if one adversarial attacks a sample towards a dummy class? I cannot really see how Eqn. (3) could justify this claim.
- "Gradients with greater variance": This part is even more confusing.
  - First question: Variance over what? Where does the randomness come from? If the model is fixed, and the data points are fixed, then the gradient is fixed. Does the randomness come from the random initialization of the model, the stochasticity in SGD, or perhaps label noise?
  - Second, why is it reasonable to assume that the gradients, if they are random, follow a Gaussian distribution that does not depend on time $t$? In some cases the Gaussian assumption could be reasonable, for example in Gaussian process NTKs. But why are they fixed throughout training? In Eqn. (4), none of $\mu_{c_n}, \sigma_{c_n}, \mu_{c_p}, \sigma_{c_p}$ depends on the training time (iteration) $t$. This makes zero sense to me.
  - Third, why is it reasonable to assume that $g_d$ is independent of $g_c$? This is required for proving Theorem 1, which uses $D(\hat{g}_c) = D(g_c) + D(g_d) + $ something else. If DuRM is implemented by adding additional linear heads on top of a shared backbone, then how could these two gradients be independent?
- Finally, I guess one very important thing is that $h_{durm}$ cannot be initialized as zero, in which case its gradient will always be zero and will never be updated (assuming that the loss function is mean squared error, for instance).

### 4. Regarding Section 2.3 about generalization analysis
First of all, which "generalization" is this section talking about? Second, I don't think this section is talking about generalization at all. This section only says that DuRM will lead to flatter local minima, but even if it is true, this does not 100% imply that the generalization would be better. Flatter minima leading to better generalization is more like a conjecture in the implicit bias community, and indeed there is some theoretical and empirical work that showed that flatter minima do improve generalization under some specific settings. But this link is not 100%. There is also prior work that shows that flatter minima do not improve generalization, such as [1].

Moreover, I cannot see how Theorem 2 could prove that DuRM leads to flatter minima. And again, this result assumes that the gradients are sampled from fixed Gaussian distributions throughout $T$ steps of training, which is not reasonable.


### 5. Regarding the experiments
First, from Table 1, I really cannot see the superiority of DuRM. And neither of Table 5 and Table 6 is compared to any SOTA method. Like I said, I feel that the only reason why DuRM could be marginally better than ERM is that it gives more degree of freedom for hyperparameter tuning. It might be true that DuRM also provides some weight-decay-like implicit regularization effect, but we can also use better optimization algorithms such as momentum SGD, Adam, or weight decay in ERM, to achieve the same effect. Thus, in my opinion, the experiments do not sufficiently justify using DuRM.

[1] Wen et al., Sharpness Minimization Algorithms Do Not Only Minimize Sharpness To Achieve Better Generalization, arXiv:2307.11007.

**Questions:**

See above.

---

### Official Review · Reviewer_TH9t · 2023-10-30

**Soundness:** 2 fair
**Presentation:** 3 good
**Contribution:** 2 fair
**Rating:** 5
**Confidence:** 3

**Summary:**

A new easy baseline, named DuRM, for improving generalisation is proposed by adding the extra output dimensions in the output space compared with ERM. The authors argue the extra output dimensions provide gradient regularisation leading to large gradient variance. Then theatrically draw the conclusion that with large gradient variance introduced by DuRM, the trained model converges to a flat loss landscape which is beneficial for generalisation ability proofed by the existing works. DuRM is empirically tested on a variety of tasks including image classification, domain generalisation, and semantic segmentation with consistent performance.

**Strengths:**

1. The proposed method performs consistently on a variety of tasks, such as image classification, domain generalisation, and semantic segmentation with different architectures.
2. The paper is well-written and easy to follow the main idea. But still, some details are not clear and sometimes confusing. I will discuss them in the later sections.
3. The idea provides a novel view of the variance of gradient which is oppositive to previous variance work. For example:

Johnson R, Zhang T. Accelerating stochastic gradient descent using predictive variance reduction. Advances in neural information processing systems. 2013;26.
Balles L, Hennig P. Dissecting Adam: The sign, magnitude and variance of stochastic gradients. International Conference on Machine Learning 2018 Jul 3 (pp. 404-413). PMLR.

**Weaknesses:**

1. When taking the mean and std of both ERM and DuRM, the performance distribution overlaps a lot. Then the improvement is not significant.
2. The intuition of the theoretical results is not very presented.
3. Hyper parameter effect is not well discussed in the submission. Not sure how it affects as the improvement is not significant. DomainBed [1] eliminates the effects of hyperparameter settings by humans.

[1] Gulrajani I, Lopez-Paz D. In search of lost domain generalization. arXiv preprint arXiv:2007.01434. 2020 Jul 2.

**Questions:**

1. How the softmax is applied to the output? If the dummy class is included for the prediction normalisation, it works like a temperature to control the training dynamic.
2. In Eq.8, the author claims if Eq.8 is satisfied DuRM can jump out of the local minimum described in Eq.7. However, it is the general case in the scholastic optimisation and it does not mean with a large gradient, jumping out of the current local minimum will end up in a better minimum. It is not clear to me how the large variance leads more flat loss region due to the reason that a large variance produces a small gradient at T.
3. By reading [2], I cannot find a similar method to empirically evaluate the local flatness of the converged point used in the submission. Usually, the eigenvalue analysis is used instead. Also when in the flat region, the gradient should be small. Can the author explain more about the empirical evidence to flat local minima? Also [2] claims the gradient directions matter and that each gradient contributes to the whole measurement.

[1] Liu Z, Xu Z, Jin J, Shen Z, Darrell T. Dropout Reduces Underfitting. arXiv preprint arXiv:2303.01500. 2023 Mar 2.

---

### Official Review · Reviewer_krmC · 2023-11-02

**Soundness:** 2 fair
**Presentation:** 2 fair
**Contribution:** 3 good
**Rating:** 5
**Confidence:** 3

**Summary:**

This paper proposes dummy risk minimization (DuRM) to improve generalization. DuRM adds some dummy classes to model predictions, i.e., if it's a K-class classification problem, DuRM will make it (K+d)-class classification problem. These ground truth labels for these d dummy classes are always set to 0.

Authors argue the DuRM reaches flatter minima due to an increase in the gradient variance -- they provide theoretical and empirical support for these arguments. DuRM is evaluated on classification & segmentation tasks on several datasets and architectures, demonstrating improved generalization in most experiments. DuRM also performs better on OOD, adversarial examples, and long-tailed datasets. Overall, DuRM is proposed as a regularization method and consistently improves test performance when incorporated into existing methods. However, the improvements are minor in terms of numbers.

**Strengths:**

- The paper presents lots of empirical experiments & evidence to support the proposed method.
- Authors considered diverse tasks, datasets, architectures, and settings and demonstrated DuRM outperforms in most cases.
- DuRM is demonstrated to improve over other regularization methods, and the approach is straightforward to incorporate into existing training pipelines. This makes it practically appealing.

Overall, the method is simple, and the authors presented a holistic analysis of their approach. They probed the method and presented interesting results, which are commendable, especially sec 3.2, 3.3, and 3.4.

**Weaknesses:**

- The author tried to explain the behavior of DuRM with some theory, but the assumptions are weak and unrealistic, and the arguments are unclear.
  - Eq 4: $p_c$ is assumed to be a mixture of Gaussian distribution. However, $0 \leq p_c \leq 1$ and assuming Gaussian distribution allow non-zero probabilities to negative or more than 1 confidence.
  - Similarly, for Thm 2, $g$ is assumed to be Gaussian instead of a mixture of Gaussian. The justification in Footnote 1 is unclear. What is implied by "..mean values of two sub-Gaussian distributions are aligned with each other.." is unclear. Moreover, if we are substituting both the actual distributions with a bound, can it be guaranteed that minimum statistics computation will hold?

- There is very little help about how many dummy classes are optimal from theory or empirical results.

- The error bars are overlapping in several results, suggesting the empirical results may not be significant. This is particularly true for all the regularization experiments (Table 7)

------
### Minor non-technical issues
I request the authors to review the language and grammar of the paper. Some instances are below (but not an exhaustive list):
- Section 1
	- ".... more generalized model state" -> what is a more generalized model state? Did authors intend "... model state that generalizes better"?
	- "... new baselines throught the..." -> new baselines throughout the
- Eq 2. why are there two $h_{erm} (x)$ functions in loss function? Perhaps the authors meant $h_{durm} (x)$.
- Def 2: iif -> iff
- MMSegmentaion -> MMSegmentaion (section 3.1)
- I could not understand footnote 2.

**Questions:**

1. Unclear theoretical arguments:
  - Is there a typo in Eq 8? Doesn't the first inequality imply the second? (assuming $\epsilon > 0$ )
  - Para before Eq 9: what does "...scaling up inequality..." mean here?

2. Section 2 "Empirical evidence to flat local minima":
  - Please check the notation -- using $||w||^2$ for the gradient norm is confusing.
  - How does a higher gradient norm ensure more flatness? Shouldn't the norm be close to 0 near minima?
  - Fig 3 (b), why does the gradient norm increase with epochs? Should it not decrease as we train more and approach minima?

3. Section 3.4: What is gradient transport randomness perspective?

4. Does Figure 3b, 4c, and 4d show test set statistics or training batch stats?

5. Authors argued that DuRM can be seen as increasing the variance of gradients. How would DuRM compare to a regularization technique like dropout?